# Low pressure amide hydrogenation enabled by magnetocatalysis

Sheng-Hsiang Lin [1,2,3], Sihana Ahmedi[1,2,3], Aaron Kretschmer[1],
Carlotta Campalani[1], Yves Kayser[1], Liqun Kang [1], Serena DeBeer [1],
Walter Leitner [1,2] & Alexis Bordet [1] ✉

The catalytic hydrogenation of amides with molecular hydrogen ($H_2$) is an appealing route for the synthesis of valuable amines entering in the preparation of countless organic compounds. Running effective amide hydrogenation under mild $H_2$ pressures is challenging although desirable to preclude the need for specialized high-pressure technologies in research and industry. Here we show that magnetocatalysis with standard supported catalysts enables unprecedented amide hydrogenation at mild conditions. Widely available and commercial platinum on alumina ($Pt/Al_2O_3$) was functionalized with iron carbide nanoparticles (ICNPs) to allow for localized and rapid magnetic induction heating resulting in the activation of neighboring Pt sites by thermal energy transfer. Exposure of the $ICNPs@Pt/Al_2O_3$ catalyst to an alternating current magnetic field enables highly active and selective hydrogenation of a range of amides at a reactor temperature of 150 °C under 3 bar or even ambient pressure of $H_2$. $ICNPs@Pt/Al_2O_3$ reacts adaptively to fluctuations in electricity supply mimicking the use of intermittent renewable energy sources. This work may pave the way toward a greatly enhanced practicability of amide hydrogenation at the laboratory and production scales, and demonstrates more generally the broad potential of the emerging field of magnetocatalysis for synthetic chemistry.

The reduction of amides to amines is a key transformation for the chemical industry and research as it produces pivotal building blocks used for the preparation of agrochemicals[1], polymers[2], dyes[3], and pharmaceuticals[4]. Traditional methods rely on the use of (over) stoichiometric reductants such as $LiAlH_4$, DIBAL, Red-Al, hydrosilanes and hydroboranes under mild conditions[5,6], making them economically and ecologically unfavourable[7,8]. The catalytic hydrogenation of amides using molecular hydrogen ($H_2$) is considered a promising alternative to produce amines in a sustainable manner following the principles of Green Chemistry[9]. This synthetic approach to amines was highlighted and prioritized by the ACS Green Chemistry Institute Pharmaceutical Roundtable as one of the highly desirable transformations[10]. The C=O bond in amides is one of the most difficult to hydrogenate among carbonyl functionalities (Fig. 1a)[11], however, hindering the development of this approach despite its obvious potential for application. As a result, significant efforts from the research community have been dedicated to the exploration of new catalysts and catalytic technologies enabling amide hydrogenation[7–9,12].

Homogeneous catalysts based on molecular organometallic catalysts with specific ligand frameworks and appropriate additives were reported to hydrogenate amides to amines (C–O bond cleavage) or to alcohols (C–N bond cleavage) at elevated temperatures and $H_2$ pressures (100–200 °C, 10–80 bar $H_2$)[9,13–16]. Solid heterogeneous catalysts have attracted attention for amide hydrogenation for a long time since they facilitate catalyst recycling and product isolation, eliminate the

[1]Max Planck Institute for Chemical Energy Conversion, Mülheim an der Ruhr, Germany. [2]Institute of Technical and Macromolecular Chemistry, RWTH Aachen University, Aachen, Germany. [3]These authors contributed equally: Sheng-Hsiang Lin, Sihana Ahmedi. ✉e-mail: alexis.bordet@cec.mpg.de

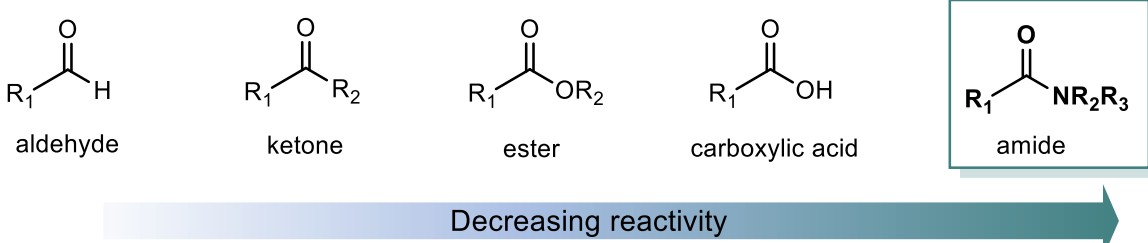

## a. C=O bonds reactivity order

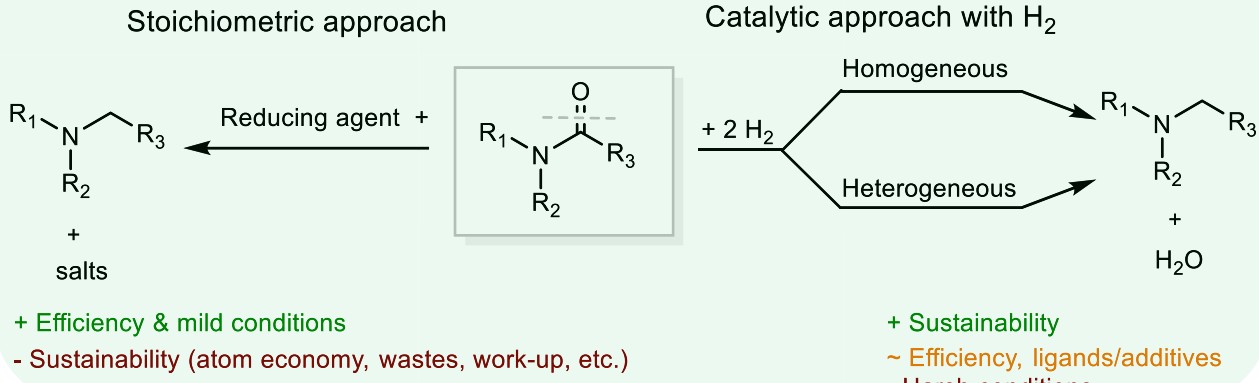

## b. Current approaches to amide hydrogenation

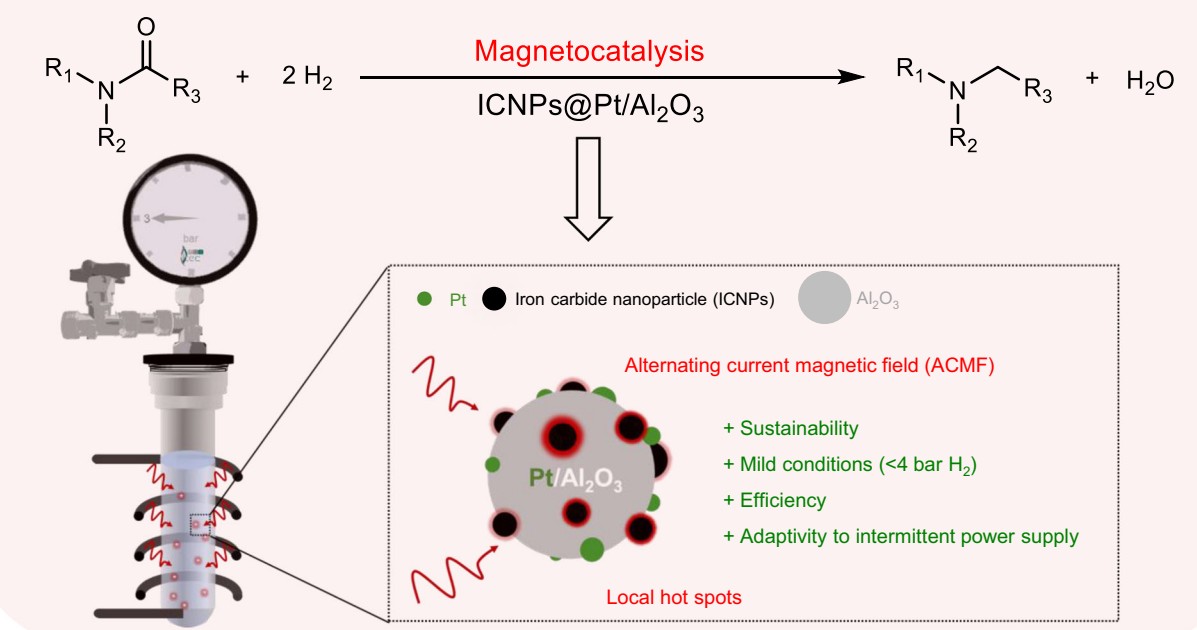

## c. This work

**Fig. 1 | Motivation and objectives of this study. a** Hydrogenation difficulty of C=O bonds in various functionalities. **b** Current approaches to amide hydrogenation. **c** the approach proposed in this work.

need for costly ligands and additives, and enable implementation into continuous flow processes toward practical application[7,17–19]. The past two decades witnessed promising advances in the application of heterogeneous catalysts for amide hydrogenation (Table S1)[12,20–24], in particular through the development of supported Pt NPs catalysts.

However, reaction conditions remain demanding (150–300 °C, 50–900 bar $H_2$) and/or associated with limited catalytic activity and stability. Amide hydrogenation with $H_2$ pressures below 5 bar would be greatly beneficial from a practical and safety point of view obviating the need for specialized high-pressure technologies on laboratory or

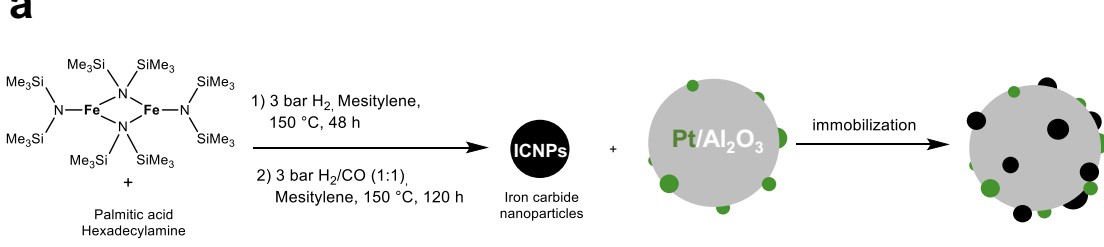

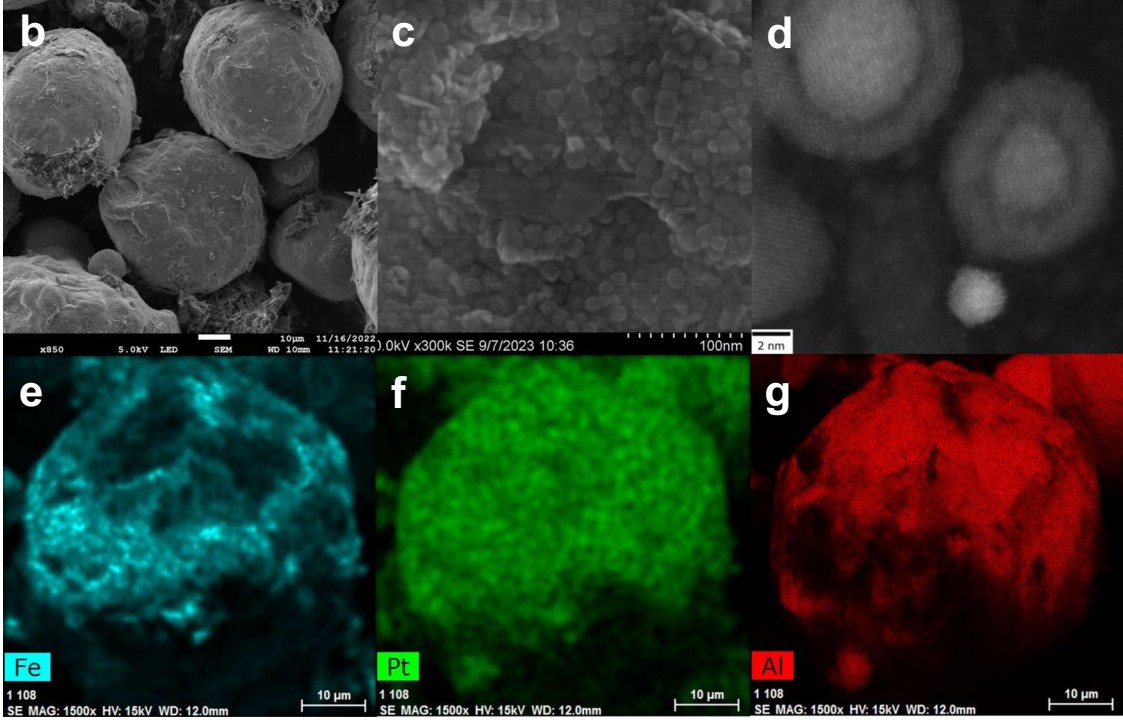

**Fig. 2 | Synthesis and characterization of ICNPs@Pt/Al₂O₃. a** Synthetic procedure for the ICNPs@Pt/Al₂O₃ catalyst. **b–g** Characterization of ICNPs@Pt/Al₂O₃ by electron microscopy including (**b**, **c**) SEM images at different magnifications showing 15 nm ICNPs on micrometer-size Pt/Al₂O₃ particles; **d** STEM-HAADF picture showing ICNPs (core-shell structure) in vicinity of ca. 2 nm Pt NPs (bright white NPs); **e–g** SEM-EDX elemental mapping of (**e**) Fe, (**f**) Pt, and (**g**) Al using Fe Kα, Pt Lα, and Al Kα.

production scale[25]. Two studies reported so far amide hydrogenation under mild conditions using specifically designed bimetallic PdRe/graphite[22] (5 bar $H_2$, 160 °C) and PtV/hydroxyapatite[24] (1–5 bar $H_2$, 25–70 °C) catalysts, although application of long reaction times (20–48 h) under barely catalytic conditions (12–40 mol% catalyst) was required. Thus, the development of new sustainable and practical catalytic technologies enabling efficient amide hydrogenation with $H_2$ under mild conditions is greatly desirable (Fig. 1b).

In the past decade, few groups including ours have pioneered the application of magnetic induction in catalysis[26,27], showing the potential of rapid, localized, and energy efficient catalyst heating to intensify catalytic processes and perform reactions such as CO and $CO_2$ hydrogenation[28,29], ketone hydro(deoxy)genation[30,31], and aldehyde hydro(deoxy)genation[32–34]. While the typical catalyst design for induction-based magnetocatalysis relied so far mainly on complex metal-doped magnetic NPs[28,31–33], we recently showed the possibility to functionalize a bulk metal catalyst ($Cu_2Cr_2O_5$) with magnetic NPs, conferring magnetic induction heating properties[30]. This allowed running a model ketone hydrogenation reaction by magnetocatalysis to demonstrate the catalyst's adaptivity to intermittent power supply.

In the present study, we set our goal to explore the broad potential of the emerging field of magnetocatalysis for synthetic chemistry by studying low pressure amide hydrogenation as an academically and industrially relevant transformation, using standard supported catalysts as versatile and widely available platforms for our catalyst design (Fig. 1c).

## Results and discussion

The major requirement for catalyst design was to combine the potential activity of widely available standard supported metal catalysts for amide hydrogenation with excellent magnetic properties for efficient heating upon exposure to magnetic induction. Thus, commercial Pt/Al₂O₃ was selected for its proven activity for amide hydrogenation at elevated temperature and pressure (180 °C and 50 bar of $H_2$)[23], and functionalized with magnetic iron carbide nanoparticles (ICNPs)[29].

ICNPs with excellent heating power under ACMF (specific absorption rate SAR of ca. 3000 W g⁻¹ at 100 kHz and 47 mT) were prepared according to an organometallic approach previously reported by some of us[29]. This involved the synthesis of Fe(0) nanoparticles (12.5 nm) from {Fe[N(SiMe₃)₂]₂}₂, followed by their carbidization under syngas (CO/H₂, 1:1 ratio) to give the ICNPs (Fe₂.₂C@Fe₅C₂, core@shell structure, 15 nm) (Fig. 2a, see SI for details). ICNPs were dispersed in THF and the resulting colloidal solution was used to impregnate commercial Pt/Al₂O₃ (1.0 wt% Pt) with a target loading of 28.5 wt% ICNPs, corresponding to a Fe loading of ca. 20 wt% (Fig. 2a, see SI for

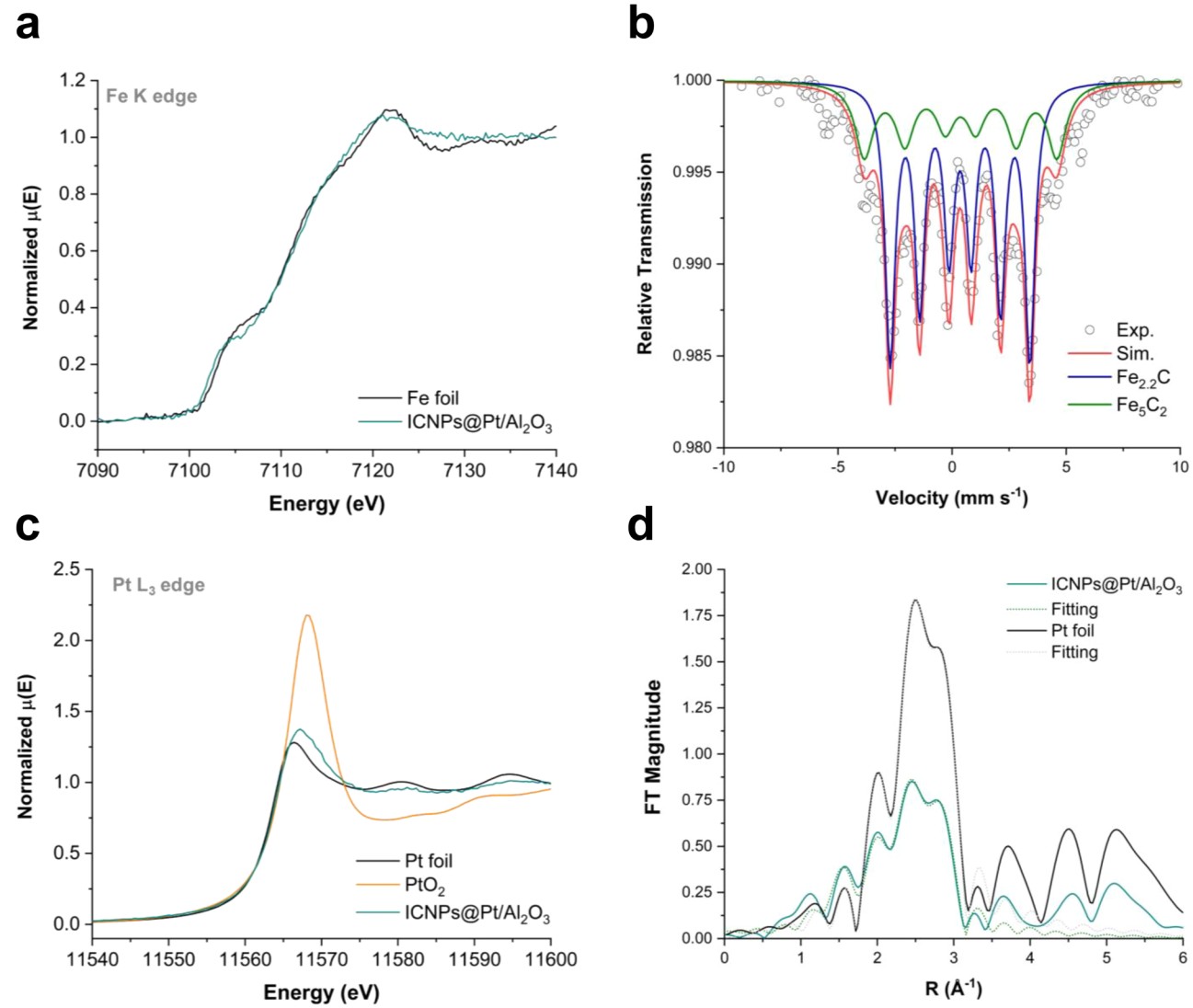

**Fig. 3 | Characterization of ICNPs@Pt/Al₂O₃. a** Fe K-edge XANES (normalized). **b** ⁵⁷Fe zero-field Mössbauer measurement at 80 K. **c** Pt L₃-edge XANES (normalized).
**d** Fourier transform magnitudes of Pt L₃-edge k²-weighted EXAFS data and corresponding fits in R-space (without phase correction).

details). The magnetic powder was dried under vacuum and treated using magnetic induction ($\mu_0 H_{max}$ = 45 mT, 350 kHz) for 1 h to anneal the ICNPs at the Al₂O₃ surface robustly preventing leaching (see SI for details).

The resulting ICNPs@Pt/Al₂O₃ material was characterized by nitrogen adsorption experiments, giving a Brunauer–Emmet–Teller (BET) specific surface area of 92.6 m² g⁻¹. This value is lower than that of starting Pt/Al₂O₃ material (155.4 m² g⁻¹), as expected due to the decoration of the material with ICNPs. Powder X-ray diffraction analysis (XRD) of ICNPs@Pt/Al₂O₃ (Fig. S1) revealed diffraction patterns characteristic of Al₂O₃ and Pt (*fcc* Pt(0)). Diffraction peaks associated with ICNPs were also visible, although some of them overlapped with Al₂O₃ and Pt signals. Elemental analysis by inductively coupled plasma optical emission spectroscopy (ICP-OES) revealed the distribution Pt = 0.6 wt%, Fe = 17.5 wt% and Al = 30.2 wt%, well in agreement with theoretical expectations (Table S2). Scanning electron microscopy (SEM) showed clearly ICNPs of expected size (ca. 15 nm) attached to the surface of quite regular spherical catalyst particles (Fig. 2b, c). Scanning transmission electron microscopy in high angle annular dark field (STEM-HAADF) at high resolution demonstrated that the ICNPs are located near Pt nanoparticles that are responsible for hydrogen activation and transfer (Fig. 2d). SEM with energy dispersive X-ray

spectroscopy (EDX) showed a fairly uniform dispersion of ICNPs and Pt across the Al₂O₃ support (Fig. 2e–g).

The magnetic properties of ICNPs@Pt/Al₂O₃ were determined using a superconducting quantum interference device (SQUID) at 300 K, showing a saturation magnetization ($M_S$) of 23.6 A m² kg⁻¹ and a coercive field ($H_C$) of 9.4 mT (Fig. S2). The electronic structure of the ICNPs@Pt/Al₂O₃ material was investigated through X-ray absorption fine structure (XAFS) analysis, focusing on the near-edge regions of the Fe K and Pt L₃ ionization thresholds (X-ray absorption near edge spectroscopy, XANES, Fig. 3). The Fe K-edge measurement showed no noticeable shift in the near-edge area as compared to a reference iron foil (7111.2 eV), and only a +0.8 eV shift in the inflection point of the rising edge (Fig. 3a). These results indicate the absence of oxidized iron and align with previous literature indicating the prevalence of an iron carbide phase[35–37]. This is consistent with ⁵⁷Fe zero-field Mössbauer spectroscopy at 80 K, which revealed the existence of two iron carbide phases (Fe₂.₂C and Fe₅C₂ in an expected 7:3 ratio) without any oxidized iron, in agreement with previous reports (Fig. 3b)[29,30,32]. The inflection point of the rising edge of the Pt L₃-edge XANES of ICNPs@Pt/Al₂O₃ is slightly shifted by +0.8 eV as compared to reference Pt foil (11567.2 eV), but is 1.0 eV lower than that of PtO₂ (11568.2 eV), pointing toward metallic Pt nanoparticles with a slightly oxidized surface (Fig. 3c). Pt L₃-

**Table 1 | Hydrogenation of 1-acetyl-3-methylpiperidine (1) using various catalytic systems and conditions**

| Heating | # | Catalyst | $P_{H2}$ (bar)[a] | $\mu_0 H$ (mT) | $T_{Reactor}$ (°C) | Conv. (%) | $Y_{1a}$ (%) |
|---|---|---|---|---|---|---|---|
| Conventional heating | 1[b] | $Pt/Al_2O_3$ | 50 | – | 200 | 43 | 43 |
| | 2 | $Pt/Al_2O_3$ | 3 | – | 200 | 2 | 2 |
| | 3 | $ICNPs@Pt/Al_2O_3$ | 3 | – | 200 | 7 | 7 |
| | 4[b] | $ICNPs@Pt/Al_2O_3$ | 3 | – | 300 | 60 | 60[c] |
| Magnetic induction heating | 5 | $ICNPs@Pt/Al_2O_3$ | 3 | 36 | 110[d] | 7 | 7 |
| | 6 | $ICNPs@Pt/Al_2O_3$ | 3 | 54 | 130[d] | 57 | 57 |
| | 7 | $ICNPs@Pt/Al_2O_3$ | 3 | 63 | 147[d] | 78 | 78 |
| | 8 | $ICNPs@Pt/Al_2O_3$ | 3 | 72 | 156[d] | >99 | >99 |
| | 9[e] | $ICNPs@Pt/Al_2O_3$ | 1 | 72 | 156[d] | >99 | >99 |
| | 10 | $ICNPs@Pt/Al_2O_3$ | 3 | 80 | 158[d] | >99 | >99 |
| | 11[f] | $ICNPs@Pt/Al_2O_3$ | 3 | 72 | 174[d] | >99 | >99 |
| | 12[g] | $ICNPs@Pt/Al_2O_3$ | 3 | 72 | – | 31 | 31 |
| | 13[h] | $ICNPs@Pt/Al_2O_3$ | 3 | 72 | – | 11 | 11 |
| | 14 | $Pt/Al_2O_3$ | 3 | 72 | r.t. | 0 | 0 |
| | 15 | ICNPs | 3 | 72 | – | 37 | 37 |
| | 16 | ICNPs + $Pt/Al_2O_3$ | 3 | 72 | – | 55 | 55 |

Reaction conditions: **1** (12.9 mg, 0.10 mmol), catalyst, decalin (0.5 mL), $H_2$ (3 bar), 4 h in a Fisher-Porter bottle. Product yields determined by GC-FID using tetradecane as the internal standard. $\mu_0 H_{max}$ (mT) magnetic field amplitude. T temperature, Conv. conversion, $Y_{1a}$ yield of **1a**.
[a]Pressure at room temperature. Increase of pressure at reaction temperature below 0.5 bar for reactions using magnetic induction heating.
[b]The reaction was performed in an autoclave (the setup illustrated in supporting information).
[c]Mass balance not closed.
[d]Determined using an infrared camera.
[e]40 h.
[f]40 wt% of ICNPs.
[g]16 wt% of ICNPs.
[h]7 wt% of ICNPs.

edge EXAFS spectra and fittings show two scattering paths in the first coordination shell: Pt–O with a coordination number (C.N.) of $1.0 \pm 0.2$ at $1.96 \pm 0.01$ Å, and Pt–Pt with a C.N. of $8.2 \pm 0.7$ at $2.75 \pm 0.01$ Å, consistent with XANES findings of metallic Pt nanoparticles with surface oxidation (Fig. 3d and Table S3). The relatively small Pt–Pt C.N. and large Debye–Waller factor (0.0060) as compared to metallic Pt (*fcc* structure) confirms the nanosized nature of the Pt components in $ICNPs@Pt/Al_2O_3$.

The amide 1-acetyl-3-methylpiperidine (**1**) was selected as a model substrate to investigate the reactivity of the $ICNPs@Pt/Al_2O_3$ catalyst. Its hydrogenation to 1-ethyl-3-methyl-piperidine (**1a**) is important for the preparation of anti-tumor agents[38]. Reactions were performed in thick-walled borosilicate glassware (Fisher-Porter bottles) under 3 bar of $H_2$ (pressure at room temperature) for 4 h with decalin as a solvent, using either conventional heating in an oil bath or magnetic induction with a commercial copper coil ($f = 350$ Hz, tunable field amplitude $\mu_0 H_{max}$, Fig. S3). Reactions at pressures >3 bar and/or temperatures >200 °C were conducted in stainless-steel autoclaves fitted with glass inlets. Using conventional heating, $Pt/Al_2O_3$ was found poorly active at 200 °C and 50 bar $H_2$ giving 1-ethyl-3-methyl-piperidine (**1a**, 43%) as the only detected product (Table 1, Entry 1), consistent with previous observations[23]. Lowering the $H_2$ pressure to 3 bar resulted in very low activity for $Pt/Al_2O_3$ and $ICNPs@Pt/Al_2O_3$ at 200 °C (Table 1, Entries 2 and 3). A moderate conversion yield of 60% **1a** could be observed when raising the temperature to 300 °C (Fig. S4), although the system pressure

increased up to 10 bar and unidentified side products resulted in a not fully closed mass balance at this extreme condition.

Using $ICNPs@Pt/Al_2O_3$ under magnetic induction at a field amplitude of 36 mT, 7% conversion to **1a** was observed (Table 1, Entry 5). A temperature of 110 °C resulting from the heat dissipated by the ICNPs was determined at the surface of the Fisher-Porter bottle by an infrared camera. Increasing the ACMF amplitude to 54 mT and 63 mT resulted in higher conversions and yields of **1a** (57% and 78%, respectively) at reactor temperatures of 130 °C and 147 °C, respectively (Table 1, Entries 6 and 7). Full conversion and quantitative yield of **1a** were observed at a moderate magnetic field amplitude of 72 mT (Table 1, Entry 7), corresponding to a global reactor temperature of 156 °C (Fig. S5) and a pressure at reaction temperature below 4 bar. Under these conditions, the local temperature at the catalyst surface was estimated to reach between 287 °C and 327 °C by placing $ICNPs@Pt/Al_2O_3$ in solvents of known boiling points and monitoring bubble formation upon ACMF exposure (Table S4). The selection of decalin as solvent appeared particularly important for magnetocatalytic reactions as activity levels were found to increase with increasing solvent boiling point (Table S4). This observation is attributed to the Leidenfrost effect, which describes the rapid vaporization of liquids at hot surfaces way above their boiling points, building an insulating vapor layer that can potentially affect mass transfer and lead to reduced reaction rates in catalysis[39]. Interestingly, using hexadecane with a boiling point of 287 °C resulted in a loss of selectivity through C–N bond cleavage and a substantially higher reactor temperature of 205 °C (Table S4). This indicates that excessive

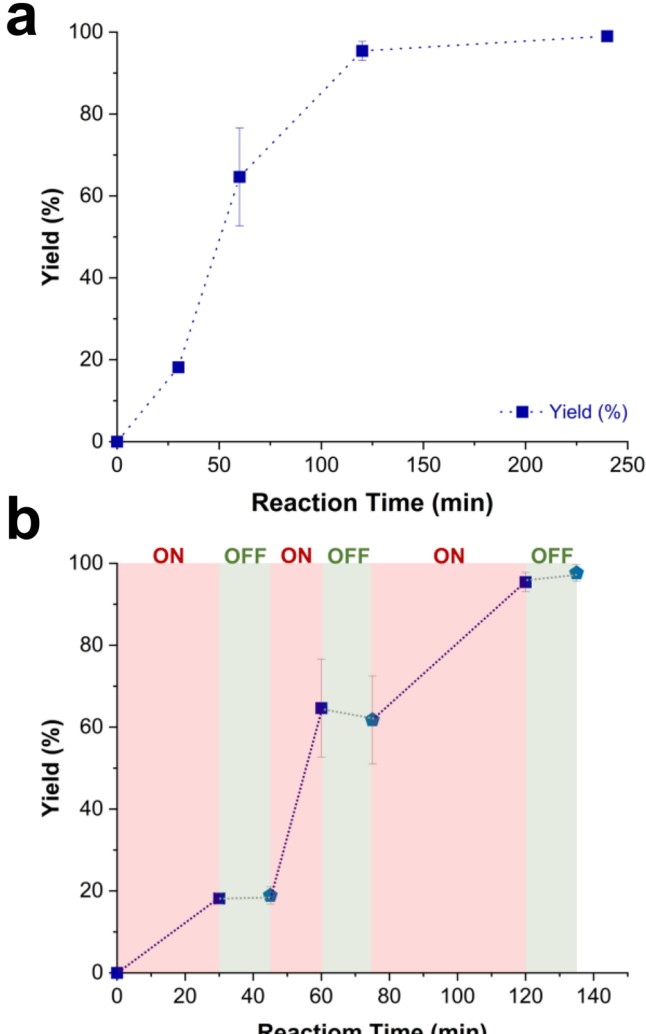

**Fig. 4 | Reaction time profiles of the magnetocatalytic hydrogenation of 1 using ICNPs@Pt/Al₂O₃. a** Time profile. **b** Time profile recorded while regularly switching ON and OFF the power supply of the magnetic induction generator (red area = power ON and green area = power OFF). <u>Reaction conditions:</u> **1** (12.9 mg, 0.10 mmol), ICNPs@Pt/Al₂O₃ (35.0 mg, 1.26 μmol Pt), decalin (0.5 mL), H₂ (3 bar), magnetic field ($\mu_0 H_{max}$ = 72 mT, 350 kHz). The product selectivity is >99%. Product yields determined by GC-FID using tetradecane as the internal standard. Data point are average values of three experiments, and error bars represent standard deviations.

temperatures are detrimental to the selectivity of the reaction. Using propylene carbonate as a green solvent proved unsuccessful due to its catalytic decomposition under these conditions. Quantitative yield of **1a** was also observed at ambient H₂ pressure, although the reaction was slower (Table 1, Entry 9). A larger ACMF amplitude of 80 mT gave as well >99% yield of **1a** at a comparable global reactor temperature (158 °C). Increasing the ICNPs loading on Pt/Al₂O₃ from 28 wt% to 40 wt% delivered quantitative yield of **1a** at a reactor temperature of 174 °C, without loss of selectivity (Table 1, Entry 11). Decreasing the ICNPs loading to 16 wt% and 7 wt% resulted in reduced catalytic activity with **1a** yields of 31% and 11%, respectively (Table 1, Entries 12 and 13), as can be expected from a decrease in the amount of heating agents. Pt/Al₂O₃ did not heat upon exposure to the ACMF and was found inactive under previously optimized conditions (Table 1, Entry 14). Pristine ICNPs were found moderately active for this transformation (37% **1a**, Table 1 Entry 15), confirming the need for the ICNPs@Pt/Al₂O₃ assembly. A physical mixture of ICNPs and Pt/Al₂O₃ gave only moderate yields of **1a** (55%,

Table 1, Entry 16). Overall, these results demonstrate that the selective heating of ICNPs@Pt/Al₂O₃ by magnetic induction enables excellent amide hydrogenation activity and selectivity under mild reaction conditions (1–3 bar H₂, T_Reactor ~ 150 °C). Such performances are strikingly out of reach for Pt/Al₂O₃ or ICNPs@Pt/Al₂O₃ catalysts heated in a conventional manner, indicating that the localized ICNPs-mediated heating of the Pt/Al₂O₃ surface at very high temperature within a colder environment is essential for the reaction to proceed smoothly at mild H₂ pressures.

In addition, the energy input provided by the magnetocatalytic process to the catalyst (conditions of Entry 8) was determined and compared to the energy input required for conventional heating at 200 °C (conditions of Entry 3). Strikingly, the magnetically activated catalyst consumed 1.4 MJ of energy to deliver >99% yield of **1a** in 4 h, while for the same reaction time conventional heating consumed 8.6 MJ and gave very poor catalytic performance (7% yield of **1a**) (See SI for details). Interestingly, 100 min were necessary for the oil bath to reach 200 °C, thereby consuming another 3.6 MJ while magnetocatalytic activation is almost instantaneous. As a result, the energy efficiency toward product formation is two orders of magnitude higher with magnetocatalysis than with conventional heating under these conditions (cf. Section "Energy Consumption Analysis" of the SI).

A time profile recorded for the conversion of **1** using ICNPs@Pt/Al₂O₃ under optimized conditions (3 bar H₂, 72 mT, 350 kHz) revealed an apparent 1st order reaction, with **1a** as the only product detected over the course of the reaction (Fig. 4a). Quantitative yield of **1a** was reached after two hours, and no further reaction or degradation of the product was observed even at prolonged reaction time.

Notably, C–N cleavage was not detected, highlighting the excellent selectivity of the ICNPs@Pt/Al₂O₃ catalyst for this substrate. Notably, turning the ACMF generator's electricity supply ON and OFF while recording a time profile resulted in the perfectly concomittent START and STOP of the catalytic reaction (Fig. 4b). For example, after 30 min the ACMF was turned OFF and no further conversion occurred. The reaction started immediately with a similar rate upon restart of the ACMF. This highlights the remarkably fast and selective heating of ICNPs@Pt/Al₂O₃ provided by magnetic induction, which are crucial features to access adaptivity to intermittent energy supply[26].

The possibility to reuse and recycle the ICNPs@Pt/Al₂O₃ catalyst was investigated using the hydrogenation of **1** (Fig. 5). For this purpose, reaction conditions were adapted and set at 72 mT and 0.5 h to ensure an incomplete conversion of **1** to **1a** allowing to monitor any change in catalytic performance. Catalytic activity and selectivity were conserved for a minimum of four cycles (Fig. 5h), with only little variations lying within experimental error. After four cycles, the BET surface area of ICNPs@Pt/Al₂O₃ increased slightly from 92.6 to 102.5 m² g⁻¹. Interestingly, an ICNPs@Pt/Al₂O₃ catalyst prepared without heat treatment under magnetic induction to anchor the ICNPs to the Pt/Al₂O₃ surface was found poorly recyclable (Table S5) presumably due to severe ICNPs leaching.

SEM and SEM-EDX revealed no significant changes in the ICNPs size nor in their distribution on the Pt/Al₂O₃ surface (Fig. 5b–g). SQUID measurement at 300 K showed a $M_S$ of 29.8 A m² kg⁻¹ and a $H_C$ of 9.5 mT (Fig. S6), very similar to that of the fresh catalyst. Fe K-edge XANES measurements of ICNPs@Pt/Al₂O₃ were found similar before and after catalysis, with only a tiny alteration ( +0.2 eV as compared to the metallic Fe) in the rising edge region for the used catalyst (Figs. 5i and S7a). This can be attributed to a trace amount of iron carbide being reduced, as can be expected under reduced conditions, and consistent with our previous findings in the hydrodeoxygenation of aldehydes[32]. Pt L₃-edge XANES spectra are nearly identical for fresh and used catalysts (Figs. 5j and S7b), and EXAFS analysis in the first coordination shell showed no change in the coordination number for the ICNPs@Pt/Al₂O₃ catalyst before and after recycling (Fig. S7c, d and

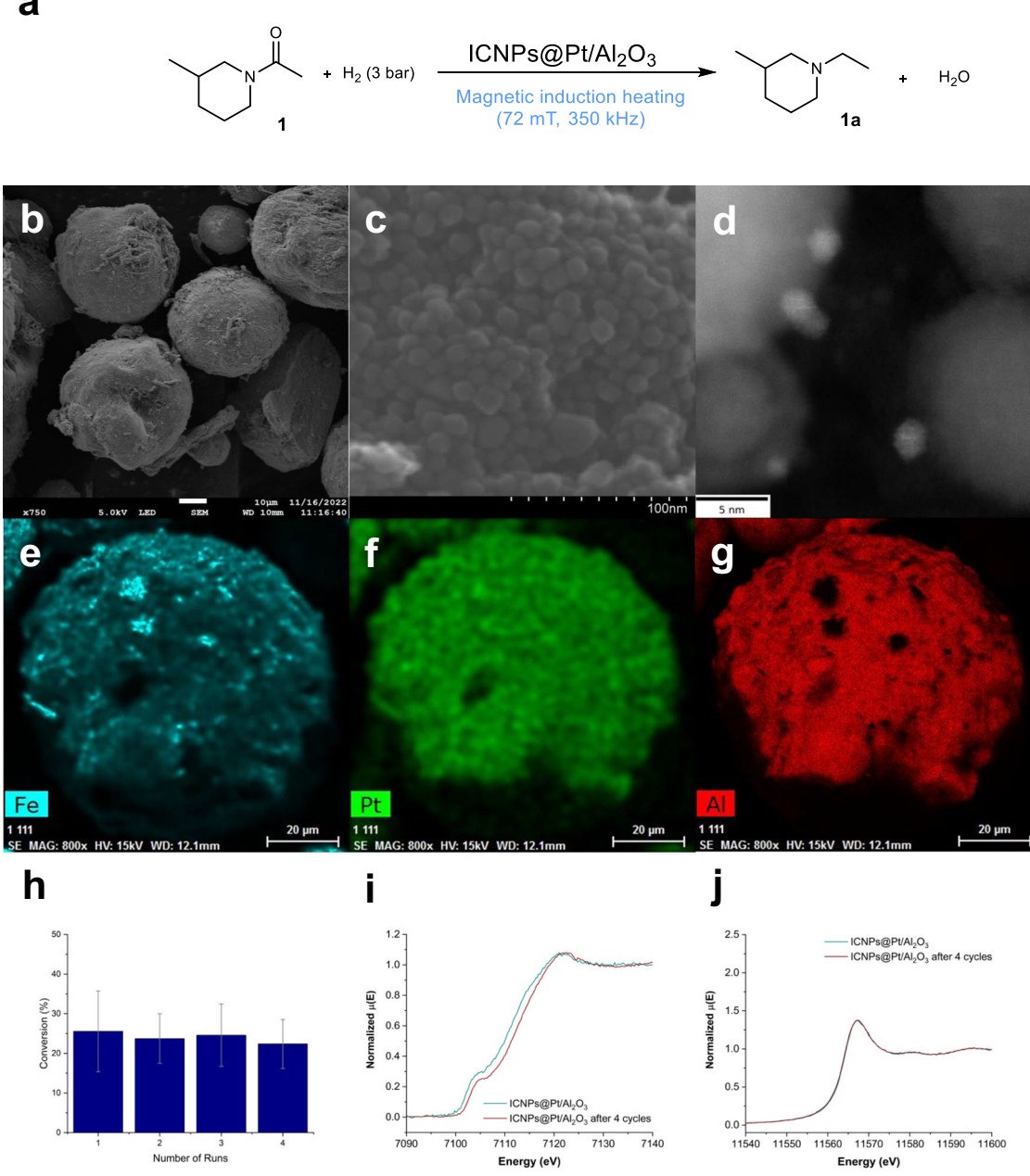

**Fig. 5 | Study of the stability of ICNPs@Pt/Al$_2$O$_3$ through recycling experiments. a** Reaction scheme. **b**–**g** Characterization of the catalyst by electron microscopy, including **b**, **c** SEM images at different magnifications, **d** STEM-HAADF, and (**e**–**g**) SEM-EDX elemental mapping, (**e**) Fe, (**f**) Pt, and (**g**) Al using Fe Kα, Pt Lα, and Al Kα. **h** Conversion of **1** and yield of **1a** over 4 cycles. **i** Fe K-edge XANES spectra (normalized). **j** Pt L$_3$-edge XANES spectra (normalized). <u>Reaction conditions:</u> **1** (12.9 mg, 0.10 mmol), ICNPs@Pt/Al$_2$O$_3$ (35.0 mg, 1.26 µmol Pt), decalin (0.5 mL), H$_2$ (3 bar), 0.5 h, magnetic field (µ$_0$H$_{max}$ = 72 mT, 350 kHz). Products yields determined by GC-FID using tetradecane as the internal standard. The product selectivity to **1a** is >99%. Data point are average values of three experiments, and error bars represent standard deviations.

Table S3). In addition, recycling experiments were also carried out under standard conditions (4 h reaction time, at incomplete and complete conversion), showing that the catalyst is robust and can provide high yields of the desired product **1a** for at least 5 consecutive cycles (20 h in total) without any make-up or regeneration (Fig. S8a, b). Elemental analysis by XRF of reaction solutions after each cycle showed no leaching of Fe nor Pt (Table S6), and changes in the Fe and Pt content of ICNPs@Pt/Al$_2$O$_3$ as determined by ICP-OES were within measurement error (Table S2).

Magnetocatalysis for low-pressure hydrogenation was explored for a scope of amides, using the ICNPs@Pt/Al$_2$O$_3$ catalyst under previously defined conditions, that were further adapted for some substrates (Table 2). Satisfyingly, the hydrogenation of tertiary heterocyclic amides (**1**–**8**) proceeded smoothly, producing the corresponding amine products in excellent yields (75–99%). Notably, **1a** (1-ethyl-3-methyl-piperidine), **2a** (1-ethylpiperidine), and **6a** (1-methylpiperidine) serve as building blocks for the synthesis of bicyclic amines that act as core intermediates in the synthesis of anti-tumor agents[39]. For substrate **3**, quantitative hydrodeoxygenation of the alcohol side group also occurred, giving a 1:1 mixture of 1-ethyl-4-methylpiperidine (**3a**) and 1-methyl-4-methylpiperidine (**3b**). The catalysts could be separated from the reaction mixtures very easily by magnetic

**Table 2 | Magnetocatalytic hydrogenation of various amides using ICNPs@Pt/Al$_2$O$_3$**

| Subs. # | Subs. | $\mu_0$H (mT) | Time (h) | Conversion (%) | Prod. # | Prod. | Yield (%) |
|---|---|---|---|---|---|---|---|
| 1 | | 72 | 4 | >99 | 1a | | >99 (95) |
| 2 | | 72 | 4 | >99 | 2a | | >99 |
| 3 | | 72 | 4 | >99 | 3a | | 50 |
| | | | | | 3b | | 50 |
| 4 | | 72 | 4 | >99 | 4a | | >99 |
| 5 | | 72 | 4 | >99 | 5a | | >99 |
| 6 | | 72 | 4 | >99 | 5a | | >99 (89) |
| 7 | | 72 | 4 | 75 | 7a | | 48 |
| | | | | | 7b | | 17 |
| | | | | | 7c | | 8 |
| 8 | | 72 | 4 | >99 | 8a | | >99 |
| 9 | | 60 | 2 | 20 | 9a | | 20 |
| | | | 4 | 48 | 9a | | 12 |
| | | | | | 9b | | 36 |
| | | | 16 | >99 | 9b | | >99 |
| 10 | | 45 | 8 | >99 | 10a | | 81 |
| | | | | | 9b | | 19 |

**Table 2 (continued) | Magnetocatalytic hydrogenation of various amides using ICNPs@Pt/Al2O3**

| Subs. # | Subs. | $\mu_0$H (mT) | Time (h) | Conversion (%) | Prod. # | Prod. | Yield (%) |
|---|---|---|---|---|---|---|---|
| 11 | | 60 | 16 | >99 | 11a | | >99 |
| 12 | | 72 | 4 | >99 | 12b | | >99 |
| | | 60 | 16 | 98 | 12a | | 95[a] |
| 13 | | 60 | 18 | >99 | 13a | | 54 |
| | | | | | 13b | | 30 |
| | | | | | 13c | | 16 |
| 14 | | 72 | 4 | 39 | 14a | | 39 |
| 15 | | 72 | 4 | >99 | 15a | | >99 |
| 16 | | 72 | 8 | >99 | 15a | | >99 |
| 17 | | 72 | 4 | >99 | 15a | | >99 |

Reaction conditions: **1** (12.9 mg, 0.10 mmol), ICNPs@Pt/Al$_2$O$_3$ (35.0 mg, 1.26 µmol Pt), decalin (0.5 mL), H$_2$ (3 bar). GC Products yields were determined relative to an internal standard of tetradecane and isolated yields are given in parentheses. For isolation: 0.4 mmol of substrate and 16 h. For substrate **11**: mixture of dioxane and decalin as solvent to ensure good substrate solubility.
[a]Rest is 3% tetrahydroindole.

separation greatly facilitating work-up and product isolation. Following this protocol, products **1a** and **5a** were isolated in excellent yields (95% and 89%, respectively) demonstrating the practicability of this approach. Hydrogenation of N-methyl pyrrolidone (**7**) gave a mixture of the expected 1-methylpyrrolidine (**7a**) along with partly (**7b**) and fully dehydrogenated products (**7c**, 1-methylpyrrole). The aliphatic substrate **9** could be selectively converted to the desired amine product **9a** under adapted conditions (60 mT, 2 h, >99% selectivity, 20% yield). C–N bond cleavage was observed upon extension of the reaction time, with butane as the only product detected (> 99% yield) after 16 h.

N-butylpropionamide (**10**) was converted into N-propylbutan-1-amine (**10a**) in high yield (81%) under mild conditions (45 mT), showing the good potential of this approach for the conversion of aliphatic secondary amides. Acetanilide (**11**) was fully converted into cyclohexane (**11a**), indicating that C–N bond cleavage and aromatic hydrogenation occurred along amide hydrogenation. A shorter reaction time gave a mixture of products including N-ethylcyclohexanamine, dicyclohexylamine, benzene, and cyclohexane (Fig. S10). Interestingly, amide functionalities in substrates **12**–**17** were hydrogenated selectively without reduction of aromatic rings.

Hydrogenation of indolin-2-one (**12**) under standard conditions also resulted in ring-opening, giving 2-ethylaniline (**12b**) in high yield. The C–N bond cleavage activity observed for this substrate and some others presumably originates from the presence of acid sites on the Pt/$Al_2O_3$ catalyst, as evidenced by pyridine FT-IR measurements (Fig. S9). Lowering the ACMF amplitude–i.e., indirectly the catalyst temperature –to 60 mT was found beneficial to prevent ring-opening through C–N bond cleavage, resulting in 95% yield of indole after 16 h (**12a**). Interestingly, **12a** was not further hydrogenated to tetrahydroindole. A comparable behavior was observed in the hydrogenation of substrate **13**, with a lesser extent of C–N bond cleavage at mild ACMF amplitude (Table 2 Entry 13, Fig. S11) and conservation of the heterocyclic C=C bond, giving products **13a**–**13c** as valuable building blocks used for the preparation of pharmaceuticals (e.g., non-steroidal aromatase inhibitors[40]). For this substrate, olefin intermediates and the additional water released in the reaction through hydrodeoxygenation may also enhance the C–N bond cleavage activity, as previously discussed in literature[41]. Interestingly, C–N bond cleavage could be suppressed also under standard conditions by neutralizing the catalyst's acid sites through addition of hexadecylamine, as demonstrated for the hydrogenation of **12** (> 99% yield of **12a** in 8 h, Fig. S12). These results indicate that adjusting reaction parameters such as ACMF amplitude and reaction time or neutralizing the catalyst's acidity can be effective strategies to limit selectivity losses through C–N bond cleavage during magnetocatalytic amide hydrogenation. Alternatively, acid-free components may also be selected in future catalyst developments. Hydrogenation of benzylamine (**15**) occurred smoothly without any aromatic hydrogenation. Benzylamine was never observed as primary product, but rapidly converted to toluene (**15a**) by debenzylation. Full reduction to toluene (**15a**) was also observed for substrates **16** and **17**, indicating protecting group cleavage as another potential application of this technique[42,43]. The corresponding amine co-products were not detected under these conditions, presumably due to the volatility of the amine moieties (e.g., from substrate **16**), and/or to their fragmentation due to C–N bond cleavage.

In conclusion, excellent amide hydrogenation performances at mild conditions were enabled by magnetocatalysis. Standard commercial Pt/$Al_2O_3$ was functionalized with magnetically-responsive ICNPs, characterized, and applied to the hydrogenation of various amide derivatives upon exposure to magnetic induction. The very intense energy transfer generated locally by ICNPs in vicinity of the Pt NPs unlocked their excellent amide hydrogenation activity at unprecedented low $H_2$ pressures (1–3 bar) allowing the use of borosilicate glassware instead of stainless steel high pressure reactors. The

possibility to operate even at ambient $H_2$ pressure obviates the need for compression steps and enables potential direct use of low pressure $H_2$ streams such as those produced by water electrolysers. The system's adaptivity to intermittent electricity supply was evidenced.

This pioneering study demonstrates the broad potential of magnetic induction as a selective heating technology compatible with virtually any supported catalyst and enabling challenging transformations under particularly mild reactions conditions. Induction-based magnetocatalysis may open new opportunities for the practical implementation in research and industry of not only sustainable amides to amines processes, but also other synthetic transformations typically requiring demanding operating conditions.

## Methods

### Materials

All syntheses were performed under argon either by using Schlenk techniques or in a glove box. Solvents were purified through a solvent purification system (MBraun-SPS-7) or dried over activated 4 Å molecular sieves then degassed and preserved under an argon atmosphere before use. Hexadecylamine (HDA, 99%), palmitic acid (PA, 99%) and platinum on alumina material (Pt/$Al_2O_3$, 1 wt% Pt) were purchased from Sigma-Aldrich. Amides are purchased from the local suppliers (e.g., Sigma-Aldrich, abcr, Alfa Aesar) and used without further purification. The bis[bis(trimethylsilyl)amido]iron(II) {Fe[N(SiMe$_3$)$_2$]$_2$}$_2$) was obtained from Nanomeps or synthesized following the literature.

### Characterization techniques

- SEM and EDX measurements were carried out using a Hitachi S-5500.
- XRD measurements were performed on a PANalytical Empyrean diffractometer using Co Kα radiation (λ = 0.1789 nm) at 45 kV and 40 mA.
- $^{57}$Fe Mössbauer spectra were collected on a spectrometer with conventional constant acceleration of the γ source ($^{57}$Co source in Rh matrix, 1.8 GBq). The sample temperature was kept constant using a Cryogen-Free Magnet with integrated variable temperature insert for zero-field measurements. The minimum experimental linewidth was 0.24 mm·s$^{-1}$. Isomer shifts are quoted relative to α-iron at 300 K. The $^{57}$Fe Mössbauer spectra were simulated and fitted with *MX* program written by Dr. Eckhard Bill.
- Superconducting quantum interference device (SQUID) data were collected on a Quantum Design MPMS-3 SQUID magnetometer. DC susceptibility was recorded at 300 K with an applied DC field of 1 T, if not stated otherwise. The SQUID data analysis was conducted with *JulX2* program written by Dr. Eckhard Bill.
- Fe K-edge data was collected using an easyXES-100 spectrometer in transmission mode. An X-ray tube with a W anode set to 25 kV and 2 mA, a 1 mm wide entrance slit, a Ge (310) crystal in second diffraction order and a silicon drift detector were used to this end. The samples were enclosed in an in-house designed anaerobic sample cell sealed with Kapton films that served as entrance and exit windows for the X-rays. The X-ray transmission of the Kapton films was included in the reference measurement without sample. The offset on the energy scale of the spectrometer was determined using an Fe foil with a thickness of 4 μm. The energy scan was repeated 60 times with an integration time of 1 s per position in each scan. The Pt L$_3$-edge XAFS spectra for the fresh and spent catalysts were collected at the P65 beamline of PETRA III (P65 applied X-ray absorption spectroscopy) in fluorescence mode due to the low relative Pt concentration[3]. This aspect and the low flux of the source made it unconceivable to extract meaningful data in a reasonable time scale with the in-house based X-ray spectrometer. At the P65 beamline, synchrotron radiation from the 3rd harmonic radiation of an 11-period undulator and monochromatized by a Si(111) double crystal

monochromator (DCM) was used. The DCM was operated in QEXAFS mode, and the undulator energy offset to the DCM was calibrated to have the maximum photon flux. Rh coated mirrors were used for focusing and collimation. The beam size at the sample position was approx. $0.5 \times 1.0$ mm$^2$ (V × H) and the photon flux was ~$10^{11}$ photons/s (without attenuation). The incident beam intensity was monitored by an ionization chamber (4 cm length, filled with 680 mbar N$_2$ and 370 mbar Ar$_{(g)}$) and the fluorescence signal was detected by a 4-element silicon drift detector (SDD). A 3 μm thick V foil was mounted in front of the fluorescence detector in order to avoid excessively high dead time due to significant Fe Kα and Kβ fluorescence from the sample. The sample pellets were prepared inside a glovebox and mounted into in-house designed fluorescence sample cells to prevent exposure to air or moisture. The measurements were performed at room temperature. The XAFS of each sample was measured 5 times and merged to improve the signal-noise ratio. A Pt foil was measured separately as the reference for energy calibration. The energy of the incident beam was calibrated by assigning the energy of the first inflection in the first derivative XANES of Pt foil to 11564 eV.

- The Pt L$_3$-edge XAFS spectra were analyzed using the Demeter software package (including Athena and Artemis programs, version 0.9.26)[4]. Pre-edge background subtraction and post-edge normalization of the XAFS data were performed using the Athena program. A linear regression background in the range of 11481 eV to 11497 eV was determined, and a quadratic polynomial regression for post-edge normalization in the range of 11598–12306 eV was applied. The fitting of EXAFS spectra (R range: 1.2–3.2 Å, k-range: 3.0–12.3 Å$^{-1}$) was performed using the Artemis program based on scattering paths generated from FEFF6. The amplitude reduction factor S$_0^2$ is determined to be 0.785 by fitting of k$^2$-weighted R-space EXAFS of the Pt foil based on the standard crystal parameters of platinum metal (retrieved from Crystal Open Database, entry ID: 9008480), and was used as fixed parameter in the EXAFS fitting model for the catalysts.

- High resolution aberration-corrected BF-STEM and HAADF-STEM images were acquired using a probe-corrected (CEOS) JEOL ARM300CF electron microscope (instrument E02) in the electron Physical Science Imaging Center at Diamond Light Source (DLS, UK). The acceleration voltage was 80 kV and the probe size was set to 8C (spot 8) with a 30 μm probe-forming aperture (CL aperture) selected, resulting in a probe convergence semi-angle of 24.8 mrad and a beam current of 28.3 pA. The STEM camera length was set to 9.0 cm, which allowed the ADF detector to integrate the scattered electron intensity between $73.7 \pm 1.8$ and $155.4 \pm 1.8$ mrad. In addition, a 3 mm aperture was inserted for the BF imaging, corresponding to a semi-angle of $14.8 \pm 1.2$ mrad (outer angle) for the BF detector. For each sample, a small amount of dry powder was sprinkled on a 200-mesh Cu grid with lacey carbon support film. Each sample was exposed to an intense electron beam for 10–15 minutes ('beam shower') to eliminate the accumulation of carbon contamination during the STEM imaging. Gatan Microscopy Suite software was used for image data acquisition.

Product Analysis was done by GC-FID (gas chromatography coupled with flame ionization detection) on a Shimadzu GC 2030 equipped with a CP-WAX-52CB column and further by GC-MS (gas chromatography coupled with mass spectrometry) on a Shimadzu QP 2020 instrument. Product quantification was done by referencing the product peak area to the peak area of the added tetradecane standard, following internal GC calibration with the isolated products. For identification of unknown products and trace compounds GC-MS was used with its internal compound library for product identification. NMR spectra were recorded on Bruker AV-400 spectrometer. The coupling constants (J)

are given in Hertz (Hz), and the chemical shifts (δ) expressed in ppm are calibrated using deuterated solvent (CDCl$_3$ at 7.26 ppm for $^1$H NMR, and 77.2 ppm for $^{13}$C NMR). The peak patterns are indicated as follows: s = singlet; d = doublet; t = triplet; m = multiplet.

### Synthesis of Fe(0) NPs

In an Ar-filled glovebox, 0.65 mmol of PA (333.2 mg) and 0.5 mmol of HDA (241.5 mg) were independently dissolved in 10 mL mesitylene and added sequentially to a green solution of 0.5 mmol {Fe[N(SiMe$_3$)$_2$]$_2$}$_2$ (376.5 mg) in 20 mL mesitylene in a Fisher-Porter (FP) bottle. The FP bottle was then pressurized with H$_2$ (2 bar) and placed in an oil bath at 150 °C for 48 h under vigorous magnetic stirring. After 48 h, the reaction was stopped and the NPs were recovered by decantation assisted by a magnet, and washed 3 times ($3 \times 10$ mL) with toluene and 3 times ($3 \times 10$ mL) with THF. The NPs were then dried under vacuum.

### Synthesis of ICNPs

In an Ar-filled glovebox, Fe(0) NPs (50 mg, 0.45 mmol of iron) were dispersed in mesitylene (9 mL) in a FP bottle, and the mixture was pressurized with CO/H$_2$ (2 bar/2 bar) at 150 °C for 120 h. At the end of the reaction, the NPs were recovered by decantation assisted by a magnet and were washed 3 times with toluene ($3 \times 5$ mL). The NPs (ca. 75 wt% Fe) were dried under vacuum.

### Synthesis of ICNPs@Pt/Al$_2$O$_3$ (28.5 wt% ICNPs loading)

In a typical experiment, ICNPs (15.0 nm; 10.0 mg) and Pt/Al$_2$O$_3$ (25.0 mg, 1 wt% Pt) were dispersed in THF (1.0 mL) in a FP bottle. The bottle was then sealed under an argon atmosphere and subjected to sonication for 1 min. At the end of the impregnation step, a black precipitate and a clear supernatant were observed. To finish, the magnetic powder was dried under vacuum and treated using magnetic induction (μ$_0$H$_{max}$ = 45 mT, 350 kHz) for 1 h to anchor the ICNPs to the Al$_2$O$_3$ surface and prevent leaching.

### Magnetocatalytic experiments (i.e., with magnetic induction heating)

In a typical experiment, ICNPs@Pt/Al$_2$O$_3$ (35.0 mg, 1.26 μmol Pt), solvent (0.5 mL), and the substrate (0.10 mmol) were placed in a FP bottle. The FP bottle was flushed, and pressurized with the desired pressure of hydrogen (typically 3 bar). The reaction mixture was placed at the center of a copper coil at the desired magnetic field amplitude and fixed frequency of 350 kHz. Once the reaction was finished, the reactor was cooled and vented. After filtration, the reaction mixture was analyzed by GC-FID using tetradecane as the internal standard.

### Catalysis with conventional heating

In a typical experiment, the catalyst, solvent (0.5 mL), and substrate (0.10 mmol) were placed in a FP bottle. The FP bottle was flushed, and pressurized with the desired pressure of hydrogen (3 bar). The reaction mixture placed in an oil bath and the reaction performed at the desired temperature. Once the reaction was finished, the FP bottle was cooled and vented. After filtration, the reaction mixture was analyzed by GC-FID using tetradecane as the internal standard. For reactions performed at higher H$_2$ pressure (50 bar H$_2$), stainless steel autoclaves heated in aluminum heating blocks were used.

### Recycling experiments

In a typical experiment, ICNPs@Pt/Al$_2$O$_3$ (35.0 mg, 1.26 μmol Pt), decalin (0.5 mL), and 1-acetyl-3-methylpiperidine (12.9 mg, 0.10 mmol) were placed in a FP bottle. The FP bottle was flushed, and pressurized with the desired pressure of hydrogen (3 bar). The reaction mixture was placed at the center of a copper coil at 72 mT and 350 kHz for 0.5 h. Once the reaction was finished, the reactor was cooled and vented. After filtration, the reaction mixture was analyzed by GC-FID using tetradecane as the internal standard. For the next cycle, fresh

portions of the substrate (0.10 mmol) and decalin (0.5 mL) were added and the reaction mixture was performed again. This procedure was repeated for each catalyst cycle by pressurizing the Fisher-Porter bottle with 3 bar of hydrogen.

## Data availability

The XAFS data and Source Data file have been uploaded to the EDMOND repository and can be found using the following https://doi.org/10.17617/3.IQVLOW. The data (including methods, supplementary Tables and Figures) generated in this study are provided in the Supplementary Information/Source Data file. Source data are provided with this paper.

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

## Acknowledgements

The work was financially supported by the Max Planck Society and Deutsche Forschungsgemeinschaft (DFG, German Research Foundation) under Germany's Excellence Strategy—Exzellenzcluster 2186 "The Fuel Science Center" ID: 390919832. Furthermore, the authors would like to thank Elena Böhme (MPI-CEC) for her help with catalyst synthesis, Christian Feike and Philipp Manthey (MPI-CEC) for their support in the Fe K-edge XAFS measurements, Annika Gurowski, Alina Jakubowski and Justus Werkmeister (MPI-CEC) for their support with GC and GC-MS measurements. Open Access provided by the Max Planck Society. We acknowledge DESY (Hamburg, Germany), a member of the Helmholtz Association HGF, for the provision of experimental facilities. Parts of this research were carried out at PETRA III (proposal No. I-20230324), and the authors would like to thank Dr. Edmund Welter for assistance in using P65 applied XAFS beamline. The authors would like to thank Diamond Light Source for access and support in use of the electron Physical Science Imaging Centre (Instrument E02) under proposal number MG35866) that contributed to the STEM results. L.K. acknowledges Alexander von Humboldt Foundation for a postdoctoral fellowship and funding support.

## Author contributions

S.-H.L. participated to the formulation of the original idea, basic project definition and conceptual planning of the experimental work flow, performed experimental work and data interpretation and wrote the manuscript draft. S.A. participated in experimental work, data interpretation and revision of the manuscript. A.K. and C.C. participated in experimental work. Y.K., L.K., and S.D. participated to the characterization of catalysts and to the writing of the manuscript. W.L. participated to the data interpretation and writing of the manuscript. A.B. participated to the formulation of the original idea, in the basic project definition and conceptual planning of the experimental work flow, supervised experimental work and contributed to data interpretation and writing of the manuscript.

## Funding

## Competing interests

The authors declare no competing interests.
