## [Transparent Peer Review file · Nature Communications]

Low Pressure Amide Hydrogenation Enabled by Magnetocatalysis

Corresponding Author: Dr Alexis Bordet

Version 0:

Reviewer comments:

Reviewer #1

(Remarks to the Author)

Designing magneto-catalysis based on magnetocaloric effect is an important and novel solution to make reaction conditions mild for numerous industrial chemical reactions. In this work, the authors prepared platinum on alumina (Pt/Al₂O₃) functionalized with iron carbide nanoparticles for hydrogenation of amides with molecular hydrogen at unprecedented mild reaction conditions. Although reported works such as "Hydrodeoxygenation Using Magnetic Induction" (Angew. Chem. Int. Ed. 2019, 58 (33), 11306-11310.) have suggested the potential of magneto-catalysis, however, the complexity of hydrogenation of amides with molecular hydrogen can still support the value and novelty of this work.

Based on the quality and novelty, it is recommended to publish this work on Nature Communications after figuring out the following questions.

1. Figure 6 shows that there are C-N bond cleavages for substrate 3,9,11,13,14,15, which might be caused by acid catalytic sites. The selectivity about C-N bond cleavages and C-O bond cleavages is a common but crucial problem for hydrogenation of amides. Could the authors explain the possible reasons caused by the difference of selectivity between substrate 1&2 and 3, especially in consideration of the similar molecular structures.
2. Commercial platinum on alumina (Pt/Al₂O₃) might introduce acid sites, and γ -Al₂O₃ verified from XRD data in SI increases the possibility of this inference. Have the authors attempted to use supports with no significant acid site for the magneto-catalysis? Theoretically C-N bond cleavages could be alleviated.
3. Table 1 entry 12 showed "A physical mixture of ICNPs and Pt/Al₂O₃ gave only moderate yields", quoted from line 196-197. In view of no activity presented for Pt/Al₂O₃ at Table 1 entry 12, activity of ICNPs itself at same conditions with Table 1 entry 12 is needed, otherwise it will be hard to evaluate the necessity to form ICNPs@Pt/Al₂O₃.
4. Line 301-302, the conversion of acetanilide proved unsuccessful, which authors attribute to low solubility. However, for Figure 6 entry 9&10, both low catalytic activity with similar phenylamino structure (Ph-N-), could it be more likely to be due to the effect of electron transfer from benzene ring?
5. Table 1 entry 5,6,7 and 9,10 showed the effect of magnetic field amplitude & loading rate of ICNPs. But both of them are monotone positive correlated with yields. Perhaps the temperature of ICNPs@Pt/Al₂O₃ particle surface have not been fully optimized yet. Generally magnetic field amplitude or loading rate of ICNPs should be increased to a certain level so that side reaction caused by high temperature of ICNPs@Pt/Al₂O₃ particle surface decrease yield or selectivity. Then compromised key reaction conditions could be set.

Reviewer #2

(Remarks to the Author)

Bordet and coworkers show a novel approach to catalytic amide hydrogenation wherein extraordinary mild conditions are employed by means of an ingenious magneto-catalysis method. It is extremely interesting that local high temperatures (hot spots) can be obtained in the reaction mixture without reaching a nearly as high global temperature, thereby potentially both saving energy and avoiding many side reactions. Furthermore, the mild conditions are particularly exciting considering the use of heterogeneous catalysis, both in light of its superiority when comparing with existing literature and for the prospects of catalyst recycling. Hence, we find this finding interesting and relevant for the broad audience - and we highly appreciate the in-depth characterization/analysis carried out as well. Having said that, we are not convinced that it is Nature Communications material in its current state. Based on the comments below, we recommend rejection in its current state:

1. Sustainability is mentioned a couple of times. We fully appreciate this statements with respect to the mild conditions. However, how sustainable is it to employ more than 1 mol% Pt catalyst? And how much energy is actually used?
2. It would help the reader to even more appreciate the concept if a comparison of the energy usage between the magneto-catalysis and conventional heating is given.
3. We understand that the authors probably aim for demonstrating the concept with a classical hydrogenation catalyst. However, considering that the general concept has been described before, including by the authors, we do not consider the work Nature Communications material. To become Nature Communications material in our minds, we are missing a more traditional optimization process. This goes for basically all the aspects, eg catalyst (why Pt and not something more sustainable/available?), solvent (sustainability can be considered also here), applied field, etc, to reach much improved reaction conditions, scope, selectivity, recyclability, and sustainability profile.
4. We are puzzled about the solvent screening. Only four solvents are given with catalysis outcome data, and there is a huge jump in boiling point between dioxane and decalin. Why not try e.g. toluene and/or derivatives, xylenes, mesitylene, high-boiling cyclic/linear/branched alkanes, DMF, DMSO, and/or NMP? And why is there no catalysis data provided for the higher-boiling solvents given in Table S4?
5. The scope is not good. Can this be improved significantly?
6. The selectivity is also not good. Can this also be improved significantly?
7. In substrate 15, it would be nice to see what the amine part becomes (see eg substrate 5).
8. In the recycling experiment, we appreciate that the authors do not simply report 4 times full conversion as that would be a useless test that is unfortunately seen too often. However, the study nevertheless comes to us too selective with respect to the reaction conditions. It puzzles us why 4x30 min are chosen - it fits remarkably well with the time-resolved study where it is seen that the catalyst is at least active for 2 hours under the given conditions. The recycling experiment must extent well beyond the boundaries of the single-batch studies.
9. At some point the authors mention the prevention of leaching, but at a later stage leaching is observed (albeit at a minor level).
10. We are missing full disclosure of crude NMRs (not just a couple of representatives).
11. It is mentioned that "The C=O bond in amides is the most difficult to hydrogenate among carbonyl functionalities." This should be specified to "among carboxylic acid derivatives" (and still carboxylate is harder to hydrogenate).

Reviewer #3

(Remarks to the Author)

Version 1:

Reviewer comments:

Reviewer #1

(Remarks to the Author)

On the basis of the reply, it is recommended to publish this work on Nature Communications after figuring out the following questions newly raised.

1. Solving, or at least suggesting certain potential solutions by preliminary feasibility tests, the problem of selectivity of amines shall be crucial to the significance of using magneto-catalysis with magnetocaloric effects for hydrogenation of amides, especially considering that serious losses were observed at scope tests. As the question 2 asked at previous round, "Have the authors attempt to use supports with no significant acid site for the magneto-catalysis", Pt/C was attempted but failed on preparing ICNPs@Pt/C. However, except acid-free catalysts, there are other methods to change acidity, such as Al³⁺/K¹⁺ supported on Al₂O₃ by impregnation. Even if the selectivity were not be improved after decreasing acidity of ICNPs@Pt/ Al₂O₃, showing other potential solutions are still be necessary. Otherwise, publication this work on Nature Communications will be unsuitable.
2. Noticed the reply for reviewer 2, figure S8 was added to declare the stability of ICNPs@Pt/ Al₂O₃ through recycling experiments. But it's insufficient, because the conversion was set nearly 100%. Thus, excess reaction conditions might cover the loss of catalyst activity.

Reviewer #2

(Remarks to the Author)

Bordet and coworkers address all our concerns in depth and fully satisfactory. We therefore recommend publication in Nature Communications almost as is:

- 1) On page 8, 3rd last line: The authors probably do not mean "...too orders of magnitude..." but perhaps "...two orders of magnitude...".

Reviewer #3

(Remarks to the Author)

Reviewer #1 (Remarks to the Author):

Designing magneto-catalysis based on magnetocaloric effect is an important and novel solution to make reaction conditions mild for numerous industrial chemical reactions. In this work, the authors prepared platinum on alumina (Pt/Al₂O₃) functionalized with iron carbide nanoparticles for hydrogenation of amides with molecular hydrogen at unprecedented mild reaction conditions. Although reported works such as "Hydrodeoxygenation Using Magnetic Induction" (Angew. Chem. Int. Ed. 2019, 58 (33), 11306-11310.) have suggested the potential of magneto-catalysis, however, the complexity of hydrogenation of amides with molecular hydrogen can still support the value and novelty of this work.

Based on the quality and novelty, it is recommended to publish this work on Nature Communications after figuring out the following questions.

=> We thank reviewer 1 for their positive opinion about our work, as well as for their valuable questions, comments, and suggestions that helped us improve the quality of the manuscript. Our point-by-point response to their comments/questions is provided below. Resulting changes in the revised manuscript are highlighted in yellow here and in the manuscript.

1. Figure 6 shows that there are C-N bond cleavages for substrate 3,9,11,13,14,15, which might be caused by acid catalytic sites. The selectivity about C-N bond cleavages and C-O bond cleavages is a common but crucial problem for hydrogenation of amides. Could the authors explain the possible reasons caused the difference of selectivity between substrate 1&2 and 3, especially in consideration of the similar molecular structures.

and

2. Commercial platinum on alumina (Pt/Al₂O₃) might introduce acid sites, and γ -Al₂O₃ verified from XRD data in SI increases the possibility of this inference. Have the authors attempted to use supports with no significant acid site for the magneto-catalysis? Theoretically C-N bond cleavages could be alleviated.

=> As pointed out by the reviewer, some of the substrates tested underwent both C-O and C-N bond cleavages, the latter being potentially caused by the presence of acid sites on our catalyst. As we could not get any information from the commercial supplier about the potential acidity of the Pt/Al₂O₃ used, we studied the surface properties of Pt/Al₂O₃ and ICNPs@Pt/Al₂O₃ by transmission FT-IR using pyridine as a molecular probe. FT IR spectra of Pt/Al₂O₃ and ICNPs@Pt/Al₂O₃ after pyridine adsorption showed a clear band at ca. 1455 cm⁻¹, confirming the presence of Lewis acidic sites on the catalysts (see Figure S9).

Notably, using pristine ICNPs directly for the hydrogenation of substrate 3 led to low conversion (6%) and no substantial C-N bond cleavage, suggesting that the sites responsible for C-N bond cleavage are present on Pt/Al₂O₃. Following the reviewer's suggestion, we attempted to prepare an acid-free ICNPs@Pt/C catalyst using commercial Pt/C. Unfortunately, the immobilization of ICNPs on Pt/C proved challenging, with severe leaching occurring presumably due to the too weak interactions between ICNPs and the carbon surface. In the near future, we plan to test non-acidic metal oxides as support materials for Pt and ICNPs, but this will require a substantial amount of work that is outside the scope of the present study.

Finally, as rightfully pointed out by the reviewer, substrates **1**, **2**, and **3** are structurally close but C-N bond cleavage is only observed for substrate **3** under standard conditions. In case of substrate **3**, the alcohol functionality is removed, however. Thus, a possibility is that the additional water produced boosts the acidity of the Al₂O₃ support, and therefore favors C-N bond cleavage. Alternatively, the intermediate alkene most likely formed through this hydrodeoxygenation pathway (but not observed here, presumably due to very fast hydrogenation) may also favor the C-N bond cleavage through coordination to the nitrogen, as previously shown in literature [Kakiuchi et al. JACS 2009, 131, 7238–7239].

This is now commented in the manuscript:

Bottom of page 12: "In this case, C-N bond cleavage (not observed for substrates **1** and **2** under these conditions) was possibly favored by this parallel hydrodeoxygenation reaction, through the release of additional water or/and through action of the olefin intermediate, as previously discussed in literature.⁴⁰ In general, the C-N bond cleavage observed for some substrates is presumably due to the presence of acid sites on the Pt/Al₂O₃ catalyst (as evidenced by pyridine FT-IR measurements, Figure S9), and may be limited in future catalyst developments by the selection of acid-free components."

3. Table 1 entry 12 showed "A physical mixture of ICNPs and Pt/Al₂O₃ gave only moderate yields", quoted from line 196-197. In view of no activity presented for Pt/Al₂O₃ at Table 1 entry 12, activity of ICNPs itself at same conditions with Table 1 entry 12 is needed, otherwise it will be hard to evaluate the necessity to form ICNPs@Pt/Al₂O₃.

=> Following the reviewer's request, a catalytic reaction with only ICNPs as catalysts was performed, and the results are now included in Table 1. While ICNPs present some level of activity for this transformation, the conversion remains modest (37%), confirming the need for the ICNPs@Pt/Al₂O₃ assembly to observe high activity.

As a second example, and as mentioned in the response above, using ICNPs for the hydrogenation of substrate **3** led to very low conversion (6%).

The new results are provided in Table 1, Entry 15, and the manuscript was modified as follows:

Middle of page 8: "Pristine ICNPs were found moderately active for this transformation (37% **1a**, Table 1 Entry 15), confirming the need for the ICNPs@Pt/Al₂O₃ assembly."

4. line 301-302, the conversion of acetanilide proved unsuccessful, which authors attribute to low solubility. However, for Figure 6 entry 9&10, both low catalytic activity with similar phenylamino structure(Ph-N-), could it be more likely to due to the effect of electron transfer from benzene ring?

=> Following the reviewer's question, additional experiments were performed using substrate **12**, indicating a mistake in the initially reported data. Interestingly, full substrate conversion was easily reached under relatively mild conditions (60 mT), although C-N bond cleavage was also observed. The substrate is insoluble in decalin at room temperature, but solubilizes after application of the ACMF due to the increase of the global temperature.

The new results were included in Table 2, and the description was changed as follows:

Middle of page 15: “Acetanilide (**11**) was fully converted into cyclohexane (**11a**), indicating that C-N bond cleavage and aromatic hydrogenation occurred along amide hydrogenation. A shorter reaction time gave a mixture of products including N-ethylcyclohexanamine, dicyclohexylamine, benzene, and cyclohexane (Figure S10).”

5. Table 1 entry 5,6,7 and 9,10 showed the effect of magnetic field amplitude & loading rate of ICNPs. But both of them are monotone positive correlated with yields. Perhaps the temperature of ICNPs@Pt/Al₂O₃ particle surface have not been fully optimized yet. Generally magnetic field amplitude or loading rate of ICNPs should be increased to a certain level so that side reaction caused by high temperature of ICNPs@Pt/Al₂O₃ particle surface decrease yield or selectivity. Then compromised key reaction conditions could be settled.

=> Following the reviewer’s question, additional experiments were performed at different ICNPs loadings and field amplitudes. Increasing the loading from 28 wt% up to 40 wt% gave full conversion and selectivity toward the desired product **1a**, with a higher global temperature of 174 °C. No loss of selectivity was observed. Increasing the ICNPs loading further did not seem attractive with regard to ICNPs utilization and dispersion of ICNPs at the Pt/Al₂O₃ surface. Similarly, increasing the ACMF amplitude up to its maximum with our equipment (80 mT) provided quantitative yields of product **1a**, without any noticeable loss of selectivity. ACMF amplitudes below 72 mT led to incomplete conversions. Thus, intermediate values of ICNPs loading and ACMF amplitude were selected to insure excellent catalytic performance while minimizing ICNPs and energy consumption.

Interestingly, using a higher boiling solvent such as hexadecane (b.p. 287 °C) allowed reaching a higher reactor temperature under standard conditions (205 °C), which led to a loss of selectivity through C-N bond cleavage (Table S4). Thus, the reviewer is in principle right, selecting conditions leading to very high temperatures can result in selectivity losses.

New results were added in Table 1, and the discussion was updated:

Middle of page 7: “Increasing the ACMF amplitude to 54 mT and 63 mT resulted in higher conversions and yields of **1a** (57% and 78%, respectively) at reactor temperatures of 130 °C and 147 °C, respectively (Table 1, Entries 6-7).”

Bottom of page 7: “Interestingly, using hexadecane with a boiling point of 287 °C resulted in a substantially higher reactor temperature of 205 °C and a loss of selectivity through C-N bond cleavage (Table S4). This indicates that excessive temperatures are detrimental to the selectivity of the reaction. Using the green solvent propylene carbonate proved unsuccessful due to its catalytic decomposition under these conditions.”

Top of page 8: “A larger ACMF amplitude of 80 mT gave as well >99% yield of **1a** at a comparable global reactor temperature (158 °C). Increasing the ICNPs loading on Pt/Al₂O₃ from 28 wt% to 40 wt% delivered quantitative yield of **1a** at a reactor temperature of 174 °C, without loss of selectivity (Table 1, Entry 11).”

Reviewer #2 (Remarks to the Author):

Bordet and coworkers show a novel approach to catalytic amide hydrogenation wherein extraordinary mild conditions are employed by means of an ingenious magneto-catalysis method. It is extremely interesting that local high temperatures (hot spots) can be obtained in the reaction mixture without reaching a nearly as high global temperature, thereby potentially both saving energy and avoiding many side reactions. Furthermore, the mild conditions are particularly exciting considering the use of heterogeneous catalysis, both in light of its superiority when comparing with existing literature and for the prospects of catalyst recycling. Hence, we find this finding interesting and relevant for the broad audience - and we highly appreciate the in-depth characterization/analysis carried out as well. Having said that, we are not convinced that it is Nature Communications material in its current state. Based on the comments below, we recommend rejection in its current state:

=> We thank reviewer 2 and 3 for their detailed evaluation of our work, as well as for their valuable questions, comments, and suggestions that helped us improve the quality of the manuscript. Our point-by-point response to their comments/questions is provided below. Resulting changes in the revised manuscript are highlighted in yellow here and in the manuscript. We are convinced that these findings will be of interest for the broad readership of *Nature Communications*.

1. Sustainability is mentioned a couple of times. We fully appreciate this statements with respect to the mild conditions. However, how sustainable is it to employ more than 1 mol% Pt catalyst? And how much energy is actually used?

=> We appreciate the reviewer's comment about sustainability, that is of course a very important factor.

Almost all heterogeneously-catalyzed amide hydrogenation reactions are performed using at least one noble metal (Ru, Rh, Pd, Pt, and combinations) under harsh conditions. The choice of Pt for this particular study was motivated by our wish to enable easy comparison of our magnetocatalytic approach with existing systems, and to start from a standard commercial catalyst that is widely used in laboratories and industries. In addition, previous literature reports claiming amide hydrogenation under mild conditions typically used specifically designed Pt-based catalysts (e.g. PtV/HAP, Kaneda et al. *Angew. Chem.* 2017, 56, 9381-9385). In the latter case, which seems to be the state of the art for mild heterogeneous amide hydrogenation (5 bar H₂ and below), **18 mol% Pt**, **21 mol% V**, and **48 h** were required to reach high yields.

In the present case, we use ~ **1.3 mol% Pt** with reaction times of **4 h**, and with a **stable** and **recyclable** catalyst. This is thus a very substantial improvement as compared to known catalytic systems.

Nevertheless, we agree with the reviewer that the replacement of noble metals is in general highly desired in catalysis. We plan to focus on 3d metal-based catalysts for similarly challenging transformations in future studies.

The question of energy consumption/efficiency is addressed in point 2.

2. It would help the reader to even more appreciate the concept if a comparison of the energy usage between the magneto-catalysis and conventional heating is given.

=> Following the reviewer's request, we determined the amount of energy transferred to the magnetically heated catalyst, and compared it to the energy input required to activate it using conventional heating.

With magnetocatalysis (72 mT), the ICNPs on ICNPs@Pt/Al₂O₃ absorb ca. 98 W^[a] of power input and release it as thermal energy, accounting for ca. 1.4 MJ over 4 hours. Importantly, the catalyst reached its working temperature almost instantaneously. Under these conditions (72 mT, 350 kHz, 3 bar H₂, 4 h), substrate **1** was fully converted, and product **1a** was obtained in quantitative yield.

Strikingly, with conventional heating, the oil bath took 100 min to reach the target 200 °C, consuming ca. 3.6 MJ as measured by a power meter. Operating it for 4 hours at 200 °C consumed another 8.6 MJ, while catalytic performance remained poor (7% conversion, 7% **1a**).

Thus, for much lower catalytic performance, conventional heating consumed ca. 12.2 MJ of energy, while the magnetically heated catalyst consumed only 1:4 MJ (see Table below for a summary).

	Magnetic (72 mT, 350 kHz)	Conventional (200 °C)
Time to target T°C (h)	0	1.67
Reactor T°C	156	200
Reaction time (h)	4	4
Energy input to reactor (MJ)	1.4	12.2
Yield of 1a (%)	>99	7
Energy efficiency toward product formation (mmol MJ ⁻¹)	0.07	0.0006

^[a]Heat generation by ICNPs under ACMF:

35 mg ICNPs@Pt/Al₂O₃ catalyst => 7 mg Fe

SAR_{ICNPs} = ca. 4000 W/g_{Fe} at 72 mT 100 kHz => ca. 14000 W/g_{Fe} at 72 mT and 350 kHz (approximation following the Stoner-Wohlfarth model)

=> **98 W** of heat released by the ICNPs in the reactor => **98 J s⁻¹**, equivalent to **1.4 MJ** in 4 h.

^{ref} H. Kreissl, J. Jin, S.-H. Lin, D. Schüette, S. Störte, N. Levin, B. Chaudret, A. J. Vorholt, A. Bordet, W. Leitner, *Angew. Chem. Int. Ed.* **2021**, *60*, 26639.

The manuscript was modified as follows:

Bottom of page 8: In addition, the energy input provided by the magnetocatalytic process to the catalyst (conditions of Entry 8) was determined and compared to the energy input required for conventional heating at 200 °C (conditions of Entry 3). Strikingly, the magnetically activated catalyst consumed 1.4 MJ of energy to deliver >99% yield of **1a** in 4 hours, while for the same reaction time conventional heating consumed 8.6 MJ and gave very poor catalytic performance (7% yield of **1a**) (See SI for details). Interestingly, 100 min were necessary for the oil bath to reach 200 °C, thereby consuming another 3.6 MJ while magnetocatalytic activation is almost instantaneous. As a result, the energy efficiency toward product formation is too orders of

magnitude higher with magnetocatalysis than with conventional heating under these conditions (cf. Section “Energy consumption analysis” of the SI).”

A section “Energy consumption analysis” was added in the SI, page 8.

3. We understand that the authors probably aim for demonstrating the concept with a classical hydrogenation catalyst. However, considering that the general concept has been described before, including by the authors, we do not consider the work Nature Communications material. To become Nature Communications material in our minds, we are missing a more traditional optimization process. This goes for basically all the aspects, eg catalyst (why Pt and not something more sustainable/available?), solvent (sustainability can be considered also here), applied field, etc, to reach much improved reaction conditions, scope, selectivity, recyclability, and sustainability profile.

=> Following the reviewer’s comment, additional experiments were performed to provide a more exhaustive optimization process. These new results are incorporated in Table 1 (ICNPs loading, ACMF amplitude) and Table S4 (solvent screening).

In brief, decreasing the ICNPs loading on Pt/Al₂O₃ led to a decrease in catalytic performance. Increasing it from 28 wt% to 40 wt% gave identical results, with quantitative yield of the desired product without loss of selectivity due for example to excessive temperatures.

Using ACMF amplitudes below 72 mT led to incomplete conversion, while increasing them up to 80 mT (limit of our equipment) gave quantitative yield of the desired product without loss of selectivity.

Additional solvents were tested in catalysis (p-Xylene, Mesitylene, Dodecane, propylene carbonate, hexadecane), fitting well with the general trend observed (see Table S4): solvents with relatively low boiling points lead to low conversions, most likely due to the Leidenfrost effect and/or the cooling of the catalyst by solvent evaporation. Unfortunately, the use of the green solvent propylene carbonate proved unsuccessful due to its decomposition by the ICNPs@Pt/Al₂O₃ under our reaction conditions. Interestingly, full substrate conversion was observed in hexadecane (b.p. of 287 °C) with a high global temperature of 205 °C. However, the selectivity was poor in this case, with substantial C-N bond cleavage.

The discussion in the manuscript was modified as follows:

Middle of page 7: “Increasing the ACMF amplitude to 54 mT and 63 mT resulted in higher conversions and yields of **1a** (57% and 78%, respectively) at reactor temperatures of 130 °C and 147 °C, respectively (Table 1, Entries 6-7).”

Bottom of page 7: “Interestingly, using hexadecane with a boiling point of 287 °C resulted in a substantially higher reactor temperature of 205 °C and a loss of selectivity through C-N bond cleavage and (Table S4). This indicates that excessive temperatures are detrimental to the selectivity of the reaction. Using the green solvent propylene carbonate proved unsuccessful due to its catalytic decomposition under these conditions.”

Top of page 8: “A larger ACMF amplitude of 80 mT gave as well >99% yield of **1a** at a comparable global reactor temperature (158 °C). Increasing the ICNPs loading on Pt/Al₂O₃ from 28 wt% to 40

wt% delivered quantitative yield of **1a** at a reactor temperature of 174 °C, without loss of selectivity (Table 1, Entry 11)."

Middle of page 8: "Pristine ICNPs were found moderately active for this transformation (37% **1a**, Table 1 Entry 15), confirming the need for the ICNPs@Pt/Al₂O₃ assembly."

The topics of sustainability, scope, selectivity, and recyclability are addressed directly in the dedicated questions.

4. We are puzzled about the solvent screening. Only four solvents are given with catalysis outcome data, and there is a huge jump in boiling point between dioxane and decalin. Why not try e.g. toluene and/or derivatives, xylenes, mesitylene, high-boiling cyclic/linear/branched alkanes, DMF, DMSO, and/or NMP? And why is there no catalysis data provided for the higher-boiling solvents given in Table S4?

=> Following the reviewer's comment, **Table S4** was updated with catalysis data in the cases for which it was not provided, as well as with new solvents. Solvents with relatively low boiling points lead to low conversions, most likely due to the Leidenfrost effect and/or the cooling of the catalyst by solvent evaporation. Unfortunately, the use of the green solvent propylene carbonate proved unsuccessful due to its decomposition by the ICNPs@Pt/Al₂O₃ under our reaction conditions. Interestingly, full substrate conversion was observed in hexadecane (b.p. of 287 °C) with a high global temperature of 205 °C. However, the selectivity was poor in this case, with substantial C-N bond cleavage.

The discussion in the manuscript was modified as follows:

Bottom of page 7: "Interestingly, using hexadecane with a boiling point of 287 °C resulted in a substantially higher reactor temperature of 205 °C and a loss of selectivity through C-N bond cleavage and (Table S4). This indicates that excessive temperatures are detrimental to the selectivity of the reaction. Using the green solvent propylene carbonate proved unsuccessful due to its catalytic decomposition under these conditions."

5. The scope is not good. Can this be improved significantly?

and

6. The selectivity is also not good. Can this also be improved significantly?

=> Following the reviewer's comments, the scope now contains 17 substrates, for the majority of which full conversion and high selectivities are reached.

For some substrates, selectivity is hampered by the substantial C-N bond cleavage activity of our catalyst, which most probably originates from the presence of acid sites. As we could not get any information from the commercial supplier about the potential acidity of the Pt/Al₂O₃ used, we studied the surface properties of Pt/Al₂O₃ and ICNPs@Pt/Al₂O₃ by transmission FT-IR using pyridine as a molecular probe. FT IR spectra of Pt/Al₂O₃ and ICNPs@Pt/Al₂O₃ after pyridine

adsorption showed a clear band at ca. 1455 cm^{-1} , confirming the presence of Lewis acidic sites on the catalysts (see Figure S9).

In addition, using pristine ICNPs directly for the hydrogenation of substrate **3** led high global temperatures but low conversion (6%) and no substantial C-N bond cleavage, suggesting that the sites responsible for C-N bond cleavage are present on Pt/Al₂O₃. Thus, improving the intrinsic selectivity of our catalyst would require some modifications to avoid the presence of acidic sites, which is out of the scope of the present study.

This is now commented in the manuscript:

Bottom of page 12: "In this case, C-N bond cleavage (not observed for substrates **1** and **2** under identical conditions) was possibly favored by this parallel hydrodeoxygenation reaction, through the release of additional water or/and through action of the olefin intermediate, as previously discussed in literature.⁴⁰ In general, the C-N bond cleavage observed for some substrates is presumably due to the presence of acid sites on the Pt/Al₂O₃ catalyst (as evidenced by pyridine FT-IR measurements, Figure S9), and may be limited by selecting acid-free components in future catalyst developments."

Nevertheless, we could add a few substrates and improve the selectivity for some others by tuning the reaction conditions (see changes in Table 2 and corresponding description).

In addition, there was a mistake for the yield of substrate 4, which is >99% and not 50%. This has been corrected.

7. In substrate **15**, it would be nice to see what the amine part becomes.

=> Following the reviewer's comment, the case of substrate **15** (now substrate **17**) was investigated, with only alkanes being detected as co-products in GC. This indicates the occurrence of C-N bond cleavage that leads to degradation of the amine-containing moiety.

A comment was added in the manuscript:

Bottom of page 15: "The corresponding amine co-products were not detected under these conditions, presumably due to the volatility of the amine moieties (e.g. from substrate **16**), and/or to their fragmentation due to C-N bond cleavage."

8. In the recycling experiment, we appreciate that the authors do not simply report 4 times full conversion as that would be a useless test that is unfortunately seen too often. However, the study nevertheless comes to us too selective with respect to the reaction conditions. It puzzles us why 4x30 min are chosen - it fits remarkably well with the time-resolved study where it is seen that the catalyst is at least active for 2 hours under the given conditions. The recycling experiment must extend well beyond the boundaries of the single-batch studies.

=> Indeed, a reaction time of 30 min was selected for recycling experiments for practical reasons: short experiments, and incomplete conversion allowing the observation of potential changes in activity and selectivity. We would like to clarify that the objective was by no means to limit the overall duration of the recycling to some specific time profile data.

Nevertheless, we agree with the reviewer that a total time of 2 h is short, and that it may make sense to perform additional longer recycling experiments.

Thus, 5 more cycles were performed under standard conditions (72 mT, 4 h), showing that the catalyst can provide high yields of the desired product for at least 5 consecutive cycles (20 h in total) without any make-up or regeneration (Figure S8).

Figure S8. Study of the stability of ICNPs@Pt/Al₂O₃ through recycling experiments. Conversion of **1** and yield of **1a** over 5 cycles. Reaction conditions: **1** (12.9 mg, 0.10 mmol), ICNPs@Pt/Al₂O₃ (35.0 mg, 1.26 μmol Pt), decalin (0.5 mL), H₂ (3 bar), 4 h, magnetic field ($\mu_0 H_{\max} = 72$ mT, 350 kHz). Products yields determined by GC-FID using tetradecane as the internal standard. The product selectivity to **1a** is >99%.

The combination of short recycling experiments at relatively low conversion and longer reactions at high yields of desired product demonstrates the stability of the catalyst, and its potential for practical use in synthesis.

Figure S8 was added in the SI, with a description/discussion in the manuscript:

Middle of page 12: "In addition, recycling experiments were also carried out under standard conditions (4 h reaction time), showing that the catalyst can provide high yields of the desired product **1a** for at least 5 consecutive cycles (20 h in total) without any make-up or regeneration (Figure S8)."

9. At some point the authors mention the prevention of leaching, but at a later stage leaching is observed (albeit at a minor level).

=> Characterization by XRF of reaction solutions during recycling experiments showed Fe and Pt contents below the detection limit (< 0.3 ppm), indicating the absence of substantial leaching of ICNPs or Pt NPs (Table S6). This was also confirmed by ICP-OES analysis of the ICNPs@Pt/Al₂O₃ catalyst after 5 cycles of 4 h (Table S2).

Table S6. XRF analysis of reaction solutions after each cycle of 4 h under standard conditions (cf. Figure S8).

Cycle	Fe [ppm]	Pt [ppm]
1	<0.3	<0.3
2	<0.3	<0.3
3	<0.3	<0.3
4	<0.3	<0.3
5	<0.3	<0.3

Table S2. Elemental analysis of ICNPs@Pt/Al₂O₃ by inductively coupled plasma optical emission spectroscopy (ICP-OES).

	Pt (wt%)	Fe (wt%)	Al (wt%)
Theoretical content	0.7	20.0	37.8
Experimental content fresh catalyst	0.6	17.5	30.2
Experimental content after catalysis (5 cycles of 4 h)	0.55	19.7	37.4

Thus, the sentence hypothesizing a minor leaching of ICNPs was modified as follows:

“After four cycles, the BET surface area of ICNPs@Pt/Al₂O₃ increased slightly from 92.6 to 102.5 m² g⁻¹, possibly due to the minor leaching of ICNPs.”

And a comment was added after the new recycling experiments:

“Elemental analysis by XRF of reaction solutions after each cycle showed no leaching of Fe nor Pt (Table S6), and changes in the Fe and Pt content of ICNPs@Pt/Al₂O₃ as determined by ICP-OES were within measurement error (Table S2).”

10. We are missing full disclosure of crude NMRs (not just a couple of representatives).

=> Product mixtures were systematically characterized using GC-FID, and the spectra are now provided in the SI (Figure S13-S62). ¹H and ¹³C NMR spectra are provided for isolated products.

11. It is mentioned that "The C=O bond in amides is the most difficult to hydrogenate among carbonyl functionalities." This should be specified to "among carboxylic acid derivatives" (and still carboxylate is harder to hydrogenate).

=> Following the reviewer's comment, this point has been modified:

Top of page 2: “The C=O bond in amides is one of the most difficult to hydrogenate among carbonyl functionalities (Figure 1a),¹¹”

We did not add “among carboxylic acid derivatives”, as it would not have fitted well with Figure 1a that includes also aldehydes and ketones.

Reviewer #3 (Remarks to the Author):

Reviewer #1 (Remarks to the Author):

On the basis of the reply, it is recommended to publish this work on Nature Communications after figuring out the following questions newly raised.

=> We thank reviewer 1 for their positive opinion about our work, as well as for their valuable questions, comments, and suggestions that helped us improve the quality of the manuscript. Our point-by-point response to their comments/questions is provided below. Resulting changes in the revised manuscript are highlighted in yellow here and in the manuscript.

1. Solving, or at least suggesting certain potential solutions by preliminary feasibility tests, the problem of selectivity of amines shall be crucial to the significance of using magneto-catalysis with magnetocaloric effects for hydrogenation of amides, especially considering that serious losses were observed at scope tests.

As the question 2 asked at previous round, "Have the authors attempt to use supports with no significant acid site for the magneto-catalysis", Pt/C was attempted but failed on preparing ICNPs@Pt/C. However, except acid-free catalysts, there are other methods to change acidity, such as Al³⁺/K¹⁺ supported on Al₂O₃ by impregnation.

Even if the selectivity were not be improved after decreasing acidity of ICNPs@Pt/ Al₂O₃, showing other potential solutions are still be necessary. Otherwise, publication this work on Nature Communications will be unsuitable.

=> Following the reviewer's request, the C-N bond cleavage occurring during the hydrogenation of some substrates was further investigated particularly interesting substrates of the scope (e.g. substrates **12** and **13**). Interestingly, varying the ACMF amplitude resulted in a substantial modulation of the selectivity, with less C-N bond cleavage observed at 60 mT than at 72 mT. These results indicate that the C-N bond cleavage pathway is favored at high amplitudes (i.e. high catalyst temperatures), and that it can be limited to some extent by optimizing ACMF amplitude and reaction time. In addition, following the reviewer's suggestion, we attempted to neutralize the catalyst's acidity through addition of hexadecylamine (ligand used in the synthesis and stabilization of ICNPs, but also a high boiling organic base). Satisfyingly, this allowed suppressing C-N bond cleavage activity even under standard conditions (72 mT), leading to 66% yield of indole (**12a**) after 4 h, and quantitative yield after 8 h (Figure S12).

These results were added (Table 2 and Figure S11-S12), and the modulation of reaction parameters as well as the addition of neutralizing agents are now commented in the manuscript as potentially promising strategies to prevent selectivity losses through C-N bond cleavage.

The manuscript was modified as follows:

Page 15: "Hydrogenation of indolin-2-one (**12**) under standard conditions also resulted in ring-opening, giving 2-ethylaniline (**12b**) in high yield. The C-N bond cleavage activity observed for this substrate and some others presumably originates from the presence of acid sites on the Pt/Al₂O₃ catalyst, as evidenced by pyridine FT-IR measurements (Figure S9). Lowering the ACMF amplitude – i.e. indirectly the catalyst temperature – to 60 mT was found beneficial to prevent ring-opening through C-N bond cleavage, resulting in 95% yield of indole after 16 h (**12a**). Interestingly, **12a** was not further hydrogenated to tetrahydroindole. A comparable behavior was

observed in the hydrogenation of substrate **13**, with a lesser extent of C-N bond cleavage at mild ACMF amplitude (Table 2 Entry 13, Figure S11) and conservation of the heterocyclic C=C bond, giving products **13a-13c** as valuable building blocks used for the preparation of pharmaceuticals (e.g. non-steroidal aromatase inhibitors⁴⁰). For this substrate, olefin intermediates and the additional water released in the reaction through hydrodeoxygenation may also enhance the C-N bond cleavage activity, as previously discussed in literature.⁴¹ Interestingly, C-N bond cleavage could be suppressed also under standard conditions by neutralizing the catalyst's acid sites through addition of hexadecylamine, as demonstrated for the hydrogenation of **12** (>99% yield of **12a** in 8 h, Figure S12). These results indicate that adjusting reaction parameters such as ACMF amplitude and reaction time or neutralizing the catalyst's acidity can be effective strategies to limit selectivity losses through C-N bond cleavage during magnetocatalytic amide hydrogenation. Alternatively, acid-free components may also be selected in future catalyst developments.”

12		72	4	>99	12b	>99
		60	16	98	12a	95 ^a

Modified entry 12 in Table 2

Figure S12. Hydrogenation of **13** using ICNPs@Pt/Al₂O₃ at 70 mT for 4 h in the presence of hexadecylamine as an additive. Reaction conditions: **12** (12.9 mg, 0.10 mmol), ICNPs@Pt/Al₂O₃ (35.0 mg, 1.26 μmol Pt), hexadecylamine (20 mg, 0.08 mmol), decalin (0.5 mL), H₂ (3 bar). >99% selectivity to **12a**.

Thus, two strategies (including preliminary validation of their effectiveness) are now suggested in the manuscript to limit C-N bond cleavage activity and improve selectivity.

N.B. The structure of product **3b** has been corrected.

2. Noticed the reply for reviewer 2, Figure S8 was added to declare the stability of ICNPs@Pt/Al₂O₃ through recycling experiments. But it's insufficient, because the conversion was set nearly 100%. Thus, excess reaction conditions might cover the loss of catalyst activity.

=> We would like to respectfully point out that recycling experiments performed at incomplete conversion were already provided in the initial manuscript submission (Figure 5h), and that Figure S8 was added specifically to address Reviewer 2's wish for recycling experiments under harsher conditions.

Nevertheless, to eliminate any potential source of confusion, additional recycling experiments were performed, this time under standard conditions but with larger amounts of substrate **1** to ensure incomplete conversion under these conditions. The results show stable catalyst performance for 7 cycles without any sign of deactivation.

Figure S8 was modified, and the manuscript was updated as follows:

Middle of page 12: "In addition, recycling experiments were also carried out under standard conditions (4 h reaction time, at incomplete and complete conversion), showing that the catalyst is robust and can provide high yields of the desired product **1a** for at least 5 consecutive cycles (20 h in total) without any make-up or regeneration (Figure S8a-b)."

Figure S8. Study of the stability of ICNPs@Pt/Al₂O₃ through recycling experiments for the conversion of **1** to **1a** at a) incomplete and b) complete conversion. Reaction conditions: **1** (25.8 mg, 0.2 mmol in a; 12.9 mg, 0.1 mmol in b), ICNPs@Pt/Al₂O₃ (35.0 mg, 1.26 μmol Pt), decalin (0.5 mL), H₂ (3 bar), 4 h, magnetic field (μ₀H_{max} = 72 mT, 350 kHz). Products yields determined by GC-FID using tetradecane as the internal standard. The product selectivity to **1a** is >99%.

Reviewer #2 (Remarks to the Author):

Bordet and coworkers address all our concerns in depth and fully satisfactory. We therefore recommend publication in Nature Communications almost as is:

1) On page 8, 3rd last line: The authors probably do not mean "...too orders of magnitude..." but perhaps "...two orders of magnitude...".

Reviewer #3 (Remarks to the Author):

=> We thank reviewers 2 and 3 for their positive opinion about our revised manuscript, and for pointing out this typo that is now corrected.